# Voltage-gated sodium channels assemble and gate as dimers

Jérôme Clatot[1], Malcolm Hoshi[1,2], Xiaoping Wan[1], Haiyan Liu[1], Ankur Jain[3], Krekwit Shinlapawittayatorn[1,2,4], Céline Marionneau[5], Eckhard Ficker[1], Taekjip Ha [3] & Isabelle Deschênes[1,2]

Fast opening and closing of voltage-gated sodium channels are crucial for proper propagation of the action potential through excitable tissues. Unlike potassium channels, sodium channel α-subunits are believed to form functional monomers. Yet, an increasing body of literature shows inconsistency with the traditional idea of a single α-subunit functioning as a monomer. Here we demonstrate that sodium channel α-subunits not only physically interact with each other but they actually assemble, function and gate as a dimer. We identify the region involved in the dimerization and demonstrate that 14-3-3 protein mediates the coupled gating. Importantly we show conservation of this mechanism among mammalian sodium channels. Our study not only shifts conventional paradigms in regard to sodium channel assembly, structure, and function but importantly this discovery of the mechanism involved in channel dimerization and biophysical coupling could open the door to new approaches and targets to treat and/or prevent sodium channelopathies.

[1] Heart and Vascular Research Center, MetroHealth Campus, Case Western Reserve University, Cleveland 44109, USA. [2] Department of Physiology and Biophysics, Case Western Reserve University, Cleveland, OH 44106, USA. [3] Department of Physics, University of Illinois at Urbana-Champaign, Champaign 61801, USA. [4] Cardiac Electrophysiology Research and Training Center, Faculty of Medicine, Chiang Mai University, Chiang Mai 50200, Thailand. [5] L'Institut du Thorax, INSERM, CNRS, UNIV Nantes, Nantes 44007, France. Eckhard Ficker is deceased. Correspondence and requests for materials should be addressed to J.C. (email: Jerome.clatot@gmail.com) or to I.D. (email: isabelle.deschenes@case.edu)

Voltage-gated sodium channels are expressed across excitable cells where they are crucial for sharp initiation dynamics and propagation of the action potential[1]. The *SCN5A* gene encodes for the cardiac sodium channel α-subunit, Na$_v$1.5. *SCN5A* mutations have been linked to cardiac arrhythmias such as Brugada Syndrome (BrS), Long QT Syndrome type 3 (LQT3), conduction slowing, sick sinus syndrome, atrial fibrillation, and dilated cardiomyopathy. Mutations in other voltage-gated sodium channel genes have also been linked to channelopathies such as epilepsy, myotonia, erythromelalgia, and ataxia[2].

Voltage-gated sodium channel genes, *SCNXA*, encode for the four domains channel[3] unlike voltage-gated potassium channel genes, which encode for one domain of the pore assembling as a tetramer. However, characterization of several *SCN5A* mutations have led to mounting evidence questioning the stoichiometry of the cardiac sodium channel. Indeed, studies of several BrS mutations have revealed the existence of mutants displaying dominant-negative effects (DN effect)[4–6]. We and others have shown that the defects of several BrS or LQT3 *SCN5A* mutations could be rescued by different *SCN5A* polymorphisms[7–11]. Finally, we also reported the presence of atypical BrS mutations that do not present defects when expressed alone but lead to reduced current amplitudes when co-expressed with wild type (WT)[12]. Altogether, these studies strongly support an interaction between α-subunits. Therefore, established lines of evidence challenge the conventional wisdom that sodium channels exist in complexes containing a single α-subunit.

Here, we report that sodium channel α-subunits not only assemble as dimers but that this physical interaction results in coupled gating. We identify the region modulating the α-α-subunit interaction and that 14-3-3 is responsible for the biophysical coupling. Importantly, deletion of the interaction site or inhibition of 14-3-3 abolished the DN effect and the biophysical coupling between α-subunits. Significantly, we demonstrate a conserved mechanism of dimerization and biophysical coupling across human voltage-gated sodium channels. Therefore, this study shifts the conventional paradigms in regard to sodium channel assembly and function, opening new therapy targets for cardiac arrhythmias and other channelopathies caused by *SCNXA* mutations.

## Results

**Sodium channels oligomerize as a dimer.** Studies of *SCN5A* mutations linked to inherited arrhythmias increasingly suggest an oligomerization of the sodium channel α-subunits. While we have previously demonstrated that α-subunits physically interact[4], here we investigated the stoichiometry of this oligomerization using multiple approaches. We first performed protein crosslinking experiments using DSS (disuccinimidyl suberate), for 20 min at different concentrations (Fig. 1a). Western blots show that increasing concentrations of DSS concur with an increase in density of a band twice the weight of a sodium channel monomer as opposed to the channel not exposed to crosslinking where only the monomeric band was present (Fig. 1a). Transferin, a transmembrane protein not known to oligomerize was used as a negative control and only one band at the monomeric size of 77 kDa was observed (Supplementary Fig. 1A). As additional controls, we performed crosslinking experiments on cells co-

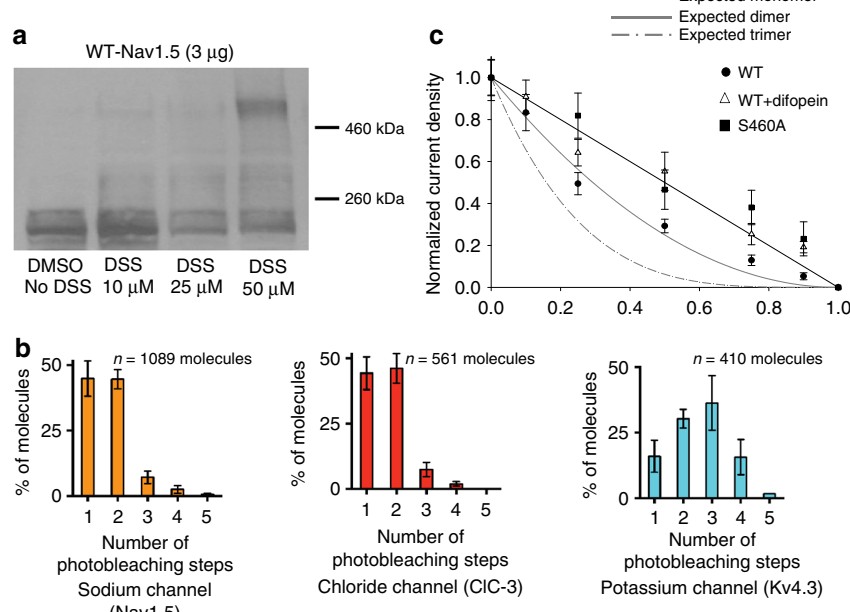

**Fig. 1** Na$_v$1.5 α-subunits form a dimer: **a** Crosslinking experiments performed with DSS in HEK293 cells expressing Nav1.5. Crosslinking was performed using DSS at different concentrations for 20 min. A band corresponding to a dimer of Na$_v$1.5 α-subunit is observed in presence of the crosslinker at higher concentrations compared to the cells where no crosslinking was performed (left DMSO no DSS). Full blots are presented in Supplementary Fig. 12. **b** GFP photobleaching steps observed through SiMPull experiments for sodium channel GFP-Na$_v$1.5, the chloride channel ClC-3-GFP known to form dimers, and the potassium channel Kv4.3-GFP known to form tetramers. The number of molecules analyzed (*n*) for each channel is indicated above each graph. Both the sodium and the chloride channels display a distribution of two photobleaching steps as would be expected for a dimeric protein whereas a distribution of four photobleaching steps was observed for the tetrameric potassium channel. **c**, Binomial analysis performed from transfected HEK293 cells using cDNA ratios 1:0, 10:1, 4:1 and 1:1 for either WT:L325R, S460A:L325R, or WT:L325R+ difopein. Current densities measured at −20 mV were normalized to the 1:0 WT or S460A currents for each ratios studied. Refer to Supplementary Table 1 for *n*. Refer to the Methods section for the equation used to obtain the theoretical curves for an expected monomer, dimer, or trimer. We can see that the current densities for the WT:L325R co-transfections decrease in a manner corresponding to an expected dimer. Importantly, when the DN mutant is co-expressed with S460A or in presence of difopein, the current density now decreases linearly as would be expected for a monomer. Data points are presented as mean ± SEM

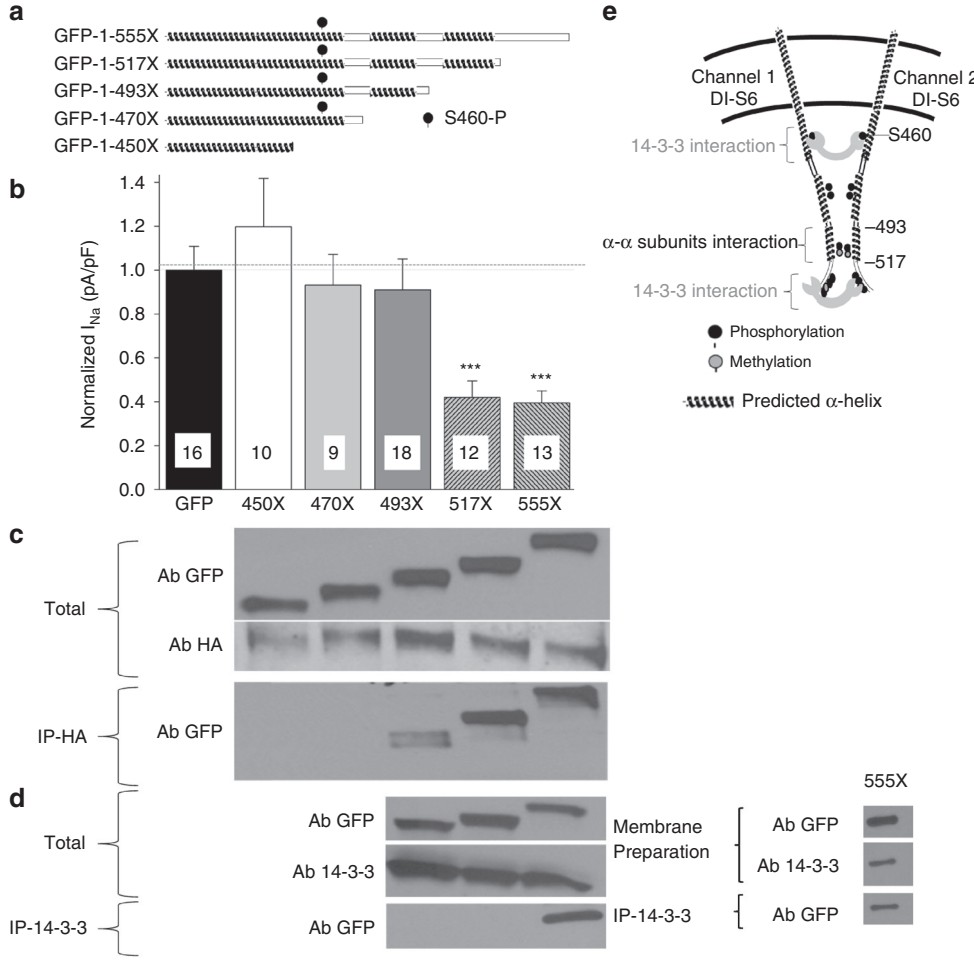

**Fig. 2** Interaction site between sodium channel α-subunits is located between amino acids 493–517. **a** Schematic representation of the different truncated GFP-Na$_v$1.5. Constructs were generated from GFP-Na$_v$1.5, where GFP is tagged to the N-terminus. Truncations were generated by inserting a stop codon at either amino acid 555, 517, 493, 470 or 450. **b** Current Density at −20 mV recorded from iCells transfected with either GFP or truncated GFP-fused Na$_v$1.5 constructs as illustrated in **a**. Data are presented as mean ± SEM, ***$p < 0.001$. The name of the construct studied is indicated below each bar of the graph. **c** Co-immunoprecipitation were performed between the full-length Na$_v$1.5 containing an HA tag and the same fragments illustrated in panel A and studied in panel B. HEK293 cells were co-transfected with HA-Na$_v$1.5 and each of the fragments separately. Two first western blots show the total proteins isolated from the transfected HEK293 cells. GFP antibody probes for the truncated sodium channel constructs and the HA antibody probes for the full-length HA-Na$_v$1.5. Co-immunoprecipitations were performed using the HA antibody and then revealed for GFP (truncated channel). The name of the truncated channel studied is illustrated above the first western. Importantly, the loss of interaction between α-subunits concurs with the loss of DN effect. Western is a representative example of four different experiments. **d** Co-immunoprecipation between truncated sodium channels illustrated in panel A and 14-3-3 were performed on total cell lysate (left) or plasma membrane preparation (right). Co-immunoprecipitations were performed using the 14-3-3 antibody and immunoprecipitations were revealed using GFP antibody to probe for interaction with the truncated sodium channels. The constructs studied in each lane are labeled at the top of panel c. In panel d, the top two westerns illustrate the total protein lysate and the last western shows the results of the co-immunoprecipitation. 14-3-3 interacts with the GFP-1-555X construct but not with the others. Westerns are representative examples of 5 different experiments. Full blots for C and D are presented in Supplementary Fig. 12. **e** Schematic representation of the region of interaction between α-subunits and between α-subunits and 14-3-3 protein in the DI-DII linker as supported from our data

expressing Na$_v$1.5 and hERG channels (Supplementary Fig. 1B). When probing the membrane for the sodium channel in presence of DSS, we obtained as in Fig. 1a a dimeric band, hence excluding a non-specific crosslinking between Na$_v$1.5 and hERG. When revealing the western blot for hERG, in presence of DSS we obtained three different bands with sizes expected for either a monomer, dimer, or tetramer as we previously reported[13]. Importantly, this experiment also provides a positive control using a protein of known oligomerization status which further validates this crosslinking approach.

Further dimerization evidence came from single-molecule pull-down (SiMPull) experiments[14]. Cell lysate expressing GFP-Na$_v$1.5 was immunoprecipitated on slides coated with the antibody of

interest (anti-GFP). GFP-Na$_v$1.5 was pulled down at a low surface density (0.05 molecules/μm$^2$) such that individual GFP-Na$_v$1.5 molecules appeared as diffraction limited spots. Fluorescence photobleaching of GFP was performed and the number of photobleaching steps for individual molecules were recorded. Nearly 50% of GFP-Na$_v$1.5 molecules bleached in 2 photobleaching steps (Fig. 1b and Supplementary Fig. 2A), while <5% of the molecules exhibited 3 or more photobleaching steps. This distribution is consistent with a dimeric stoichiometry of sodium channels, given that only ~70% of GFP is fluorescently active[14,15]. The molecules bleaching in two steps exhibited a longer fluorescence photobleaching time, as expected (Supplementary Fig. 2D). For molecules bleaching in 2 distinguishable steps, the

mean photobleaching time for the first step is $0.9 \pm 0.2$ s while that for the complete bleaching (total time till the second step) is $4.5 \pm 2.0$ s. In contrast, the mean bleaching time for the molecules bleaching in one-step only is $1.4 \pm 0.2$ s. The short photobleaching time for the first bleaching step is indeed expected, when the bleaching of two fluorophores is stochastic and independent of one another. We also used a GFP-tagged chloride channel (ClC-3) as a known and validated dimeric control, as well as the transient outward potassium channel $K_v4.3$ tagged with GFP, as a known tetrameric channel control. As expected for a dimeric protein, close to 50% of chloride channel molecules displayed 2-step photobleaching (Fig. 1b and Supplementary Fig. 2B). It is worth noting the outstanding similarity in the distribution of photobleaching steps between the known dimeric chloride channel and the sodium channel. For the $K_v4.3$ potassium channel a majority of the molecules exhibited 1 to 4 photobleaching steps, as expected for a tetrameric channel (Fig. 1b and Supplementary Fig. 2C). Our team has also previously validated a monomeric membrane protein using a mitochondrial outer membrane protein MAVS[14]. This monomeric membrane protein tagged with YFP displayed 86% single photobleaching steps which is significantly distinct from what we observed for the sodium channel. Therefore, the use of the SiMPull technique further supports the concept that sodium channels form dimers.

Our third approach to assess the stoichiometry uses a binomial analysis based on the current density obtained from varying the ratio of DN-mutants to WT as previously described for potassium channels[16]. We used two DN-mutants, L325R and R104W, which display little to no currents when expressed on their own[4,5]. HEK293 cells were co-transfected with different L325R:WT or R104W:WT cDNA ratios 0:1, 1:10, 1:4, 1:1, 4:1, 10:1, 1:0 increasing the amount of the DN-mutants cDNA. The 0:1 ratio (100% $Na_v1.5$ WT) was considered 100 percent current as opposed to the 1:0 ratio (100% L325R or R104W) in which no current was detected. Interestingly, the decrease in current density associated with the increasing amount of mutant cDNA (L325R and R104W) was not linear as would be expected if the two sodium channels were independent monomers, but instead is well fit with a binomial curve that matches a dimeric configuration (Fig. 1c). Importantly, Supplementary Fig. 3 illustrates representative raw current traces examples and full I/V curves for the different conditions demonstrating the reliability of the level of expression. In addition, we performed a control binomial analysis by co-expressing WT with the trafficking but non-conducting mutant R878C which does not produce a DN effect[4]. Strikingly, the full binomial analysis using this mutant demonstrates that the reduction in $I_{Na}$ current density now follows a linear decrease as expected for a non-DN mutant (Supplementary Fig. 3C). Hence, using these three different approaches (crosslinking, SiMPull, and binomial analysis) our data provide evidence demonstrating dimerization of sodium channel α-subunits.

**The DN effect is a result of direct suppressive effect**. It is possible that overexpression of an improperly folded protein could imbalance the protein expression system and potentially lead to a non-specific DN effect due to ribosome or translocon competition effects. To ensure that the observed DN effect is not due to translocon or ribosome competition, we measured the expression level of two endogenous soluble proteins, GAPDH and actin, and show that the level of expression of these proteins are identical in presence of the WT or the DN mutant L325R (Supplementary Fig. 4A). We also addressed the possible ribosome or translocon competition issue by measuring $I_{CaL}$ and $I_{K1}$ in cardiac myocytes expressing L325R (Supplementary Fig. 4B,C)

and by measuring $I_{to}$ and $I_{Kr}$ in HEK293 cells where we co-expressed $Na_v1.5$ WT or L325R with $K_v4.3$ or hERG channels (Supplementary Fig. 4D,E). Importantly, none of the measured current either in native cells or heterologous expression system were altered by the presence of the DN mutant L325R compared to WT $Na_v1.5$. Finally, no changes in activation or inactivation were seen in presence of the DN mutant (Supplementary Fig. 5), suggesting that the effect of the DN mutant channels is simply suppressive. Altogether, these results help exclude a ribosome or translocon hypothesis and a biophysical influence, supporting once more the specific DN mechanism caused by interaction between two sodium channel α-subunits.

**$Na_v1.5$ α-subunits interaction site**. The next step was to identify the region responsible for the α–α subunit interaction and dimerization. We previously demonstrated that two α–subunits lacking the C-terminus region are able to interact[17], suggesting that the C-terminal is not the main region of interaction. Moreover, Park et al.[6] recently described a pig transgenic model containing the E555X truncated mutation reported in a BrS patient. Their recordings show a DN effect of this mutation on endogenous sodium currents in pig cardiomyocytes, suggesting that the interaction between two channels occurs within the first 555 amino acids. Thus, we generated a N terminus GFP-fused $Na_v1.5$-E555X construct to investigate the DN effect in iPS-derived cardiomyocytes (iCells) (Fig. 2a). Importantly, the expression of E555X in iCells, exerted a DN effect on the endogenous sodium current (Fig. 2b and Supplementary Fig. 6), reproducing the observation from the pig model. We postulate based on our previous work[4] that the DN effect observed is mediated through interactions between the truncated DN mutant and the WT endogenous channel. To assess this, we performed co-immunoprecipitation in HEK293 cells where the GFP-tagged truncated 555X channel was co-expressed with the full-length HA-tagged-$Na_v1.5$. As expected for a mutant presenting with a DN effect, the E555X fragment co-immunoprecipitated with the full-length channel (Fig. 2c) suggesting that the interaction site between sodium channel α-subunits is located within the first 555 amino acids of the channel. In order to hone in on the interaction region between α-subunits, we generated different truncated sodium channels based on phyre2 modeling tridimensional structure predictions[18] and the previously reported 14-3-3 interaction site[19] (Fig. 2a, e). On the basis of the predicted α-helix and post-translational pattern[20], we created the truncated constructs 450X, 470X (before and after the previously reported 14-3-3 interaction site and the phosphoserine 460), as well as 493X and 517X in order to cut the predicted α-helixes. We investigated in parallel the DN effect and the interaction of the different fragments. Strikingly, the 450X and 470X constructs neither exerted a DN effects (Fig. 2b) nor co-immunoprecipitated with the full-length channel (Fig. 2c). The 493X truncated channel had weak interaction with the full-length channel (Fig. 2c) and this weak interaction was not sufficient to lead to a DN effect (Fig. 2b). However, 517X and 555X both interacted strongly with the channel and exerted DN-effects (Fig. 2b, c). Altogether our results strongly suggest that the main interaction site between two α-subunits resides between amino acids 493 and 517.

**DN effect abolished by 14-3-3 inhibition**. 14-3-3 has been shown to interact with the DI-II linker between amino acid 417 and 467 and 14-3-3 is known to facilitate the formation of dimers[19]. Therefore, we investigated the potential involvement of 14-3-3 in the dimerization of sodium channels by inhibiting its action: either with the 14-3-3 inhibitor difopein[21] or by mutating

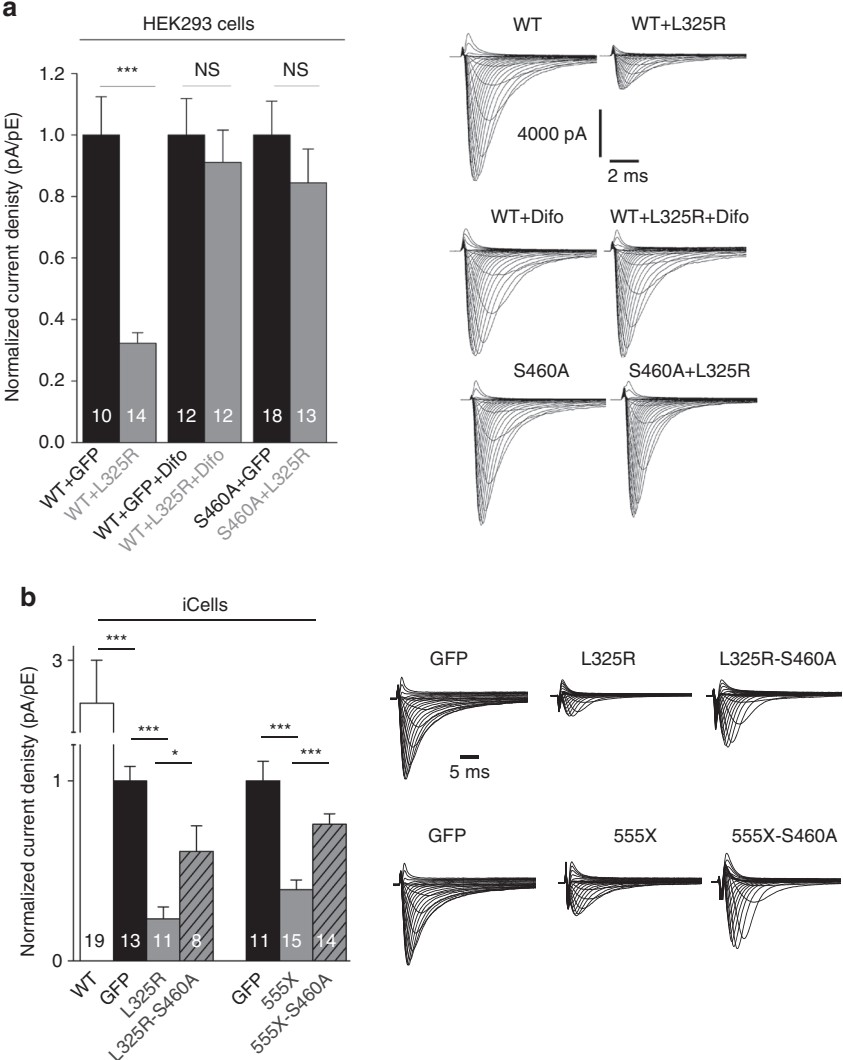

**Fig. 3** 14-3-3 inhibition abolishes DN effect. **a** Normalized current density at −20 mV recorded from transfected HEK293 cells. HEK293 cells were transfected with the different conditions illustrated at the bottom of the graph. WT corresponds to the wild-type cardiac sodium channel, L325R is the DN mutant, difopein (difo) corresponds to a 14-3-3 inhibitor and S460A is a mutant of the sodium channel where the 14-3-3 relevant phosphorylation site (S460) is mutated to alanine. Right panel displays representative family current traces for each conditions. **b** Normalized current density at −20 mV recorded from iCells transfected with different DN-mutants, GFP or WT. Transfection conditions are illustrated at the bottom of the graph. Right panel displays GFP normalized representative family current traces. n are noted on the bars, data are presented as mean ± SEM, *$p < 0.05$, ***$p < 0.001$, NS not significant

to alanine the serine S460 found within the 14-3-3 interaction site (S460A) since 14-3-3 interaction is phosphoserine dependent. Strikingly, the L325R DN effect was abolished in presence of difopein or with S460A-Na$_v$1.5 channel (Fig. 3a and Supplementary Fig. 7A). Importantly, we performed the binomial analysis from Fig. 1c and Supplementary Fig. 3 with either difopein or S460A present. The currents measured under these conditions followed a linear decrease, strongly suggesting that WT in presence of either difopein or S460A-Na$_v$1.5 channel are now behaving as monomers (Fig. 1c). Moreover, we confirmed these results with another DN mutant, R104W[4], where once again the presence of difopein abolished the DN effect exerted by this R104W mutant channel (Supplementary Fig. 7B), and currents decreased in a linear manner in presence of difopein or S460A variant on the binomial analysis study (Supplementary Fig. 3). Altogether, these results suggest that the channels are behaving independently or as monomers in presence of difopein or S460A variant.

We next confirmed that this mechanism is conserved in cardiomyocytes. To do so, we inserted the S460A variant on the L325R cDNA construct and expressed it in iCells and then measured current densities. We observed a DN effect when iCells were transfected with the L325R-Na$_v$1.5, compared to GFP transfected iCells (control) or cells transfected with the WT; however, this DN effect was significantly impaired when S460A was inserted in the construct (L325R-S460A) (Fig. 3b). We then used the 555X DN mutant previously described in a pig transgenic model[6] to assess if once again we could impair the DN effect in myocytes by inhibiting 14-3-3. Importantly, not only were we able to replicate this DN effect in myocytes when measuring current densities but we also impaired it by inserting S460A on the E555X cDNA construct (555X-S460A) (Fig. 3b). Altogether, these results suggest that 14-3-3 is implicated in the DN mechanism.

Given the implication of 14-3-3 in the DN mechanism, we then assessed interaction of 14-3-3 with the different truncated

channels to identify the 14-3-3 interaction site. Co-immunoprecipitation between Na$_v$1.5 fragments and 14-3-3 is illustrated in Fig. 2a; we observed that 14-3-3 interacts only with the highly phosphorylated/methylated region found between amino acids 517 and 555 since no bands were observed for 493X and 517X (Fig. 2d left). The co-immunoprecipitation between the channel and 14-3-3 were first performed from whole-cell lysates. In order to assess interaction at the cell surface between the two partners, we also performed co-immunoprecipitations using membrane preparations and confirmed that the interaction still occurs at the plasma membrane (Fig. 2d right). This 14-3-3 interaction region identified is downstream from the Na$_v$1.5 α-subunits interaction site (Fig. 2b, c) suggesting that 14-3-3 does not mediate the direct dimerization of sodium channels even though it is implicated in the DN mechanism (Fig. 3 and Supplementary Fig. 7).

**Sodium channels display coupled gating**. To this point, our co-immunoprecipitation did not discriminate whether the interaction occurs during channel assembly, trafficking or at the plasma membrane. To investigate α-subunits interaction at the plasma membrane, TIRF/FRET receptor photobleaching was performed in HEK293 cells co-expressing Na$_v$1.5-CFP and Na$_v$1.5-YFP channels. These experiments showed significant increase in CFP signal after YFP receptor photobleaching (Fig. 4) demonstrating that Na$_v$1.5 α-subunits exist within FRET range of each other (~10 nm) and therefore strongly suggesting that the channels also interact at the plasma membrane. A membrane-localized CFP-YFP tandem construct (Rho-pYC) was used as a positive control[22], and a potassium channel accessory subunit KCNE1-YFP co-expressed with Na$_v$1.5-CFP was used as a negative control.

This interaction at the cell surface led us to question whether this results in biophysical coupling between two channels. To investigate biophysical coupling, we co-expressed with the WT a mutant displaying defective inactivation properties, R1629Q (Fig. 5). We first studied steady-state inactivation (SSI) with this mutant which displays a large hyperpolarizing shift in SSI but yet has current densities similar to WT. If the two channels function independently, when co-expressing R1629Q with WT channel we would expect the resulting SSI to be an average of both different channels and be roughly in the middle (theoretical curve shown as dotted line in Fig. 5a). Surprisingly, the WT+ R1629Q SSI displayed in Fig. 5a, did not match the theoretical curve expected

for two independent channels, but instead behaved closer to WT suggesting that the WT could partially rescue R1629Q SSI. Given the difficulty of working with two conducting channels for which similar current density must be assumed, we further demonstrate coupled gating by co-expressing R1629Q with the trafficking competent but non-conductive R878C channel[4]. When co-expressing R1629Q with R878C, R1629Q is the only channel carrying I$_{Na}$; if the two channels were independent from each other the presence of the non-conductive R878C should not affect R1629Q biophysical properties. However, the presence of R878C channel exerted a significant shift on SSI compared to R1629Q expressed alone (Fig. 6a). In order to investigate if coupled gating between α-subunits can modulate other biophysical parameters, we measured R1629Q inactivation decay and activation when expressed alone or in presence of R878C. The presence of R878C significantly slowed down R1629Q inactivation decay (Fig. 6b), and the voltage-dependent activation was also significantly shifted (Fig. 6c, d).

Having determined that 14-3-3 inhibition abolishes the DN effect (Fig. 3 and Supplementary Fig. 7) and that 14-3-3 interacts with the channel at the cell surface (Fig. 2d), we now investigated whether 14-3-3 inhibition can also biophysically uncouple the dimers. We studied the biophysical properties of R1629Q co-expressed with WT or R878C in presence of the 14-3-3 inhibitor (Figs 5b and 6a–c right panels). Strikingly, the presence of difopein abolishes the shifts observed in presence of R878C for all biophysical parameters studied (Fig. 6a–c, right panels). Similarly, when co-expressing R1629Q+WT in presence of difopein, the resulting SSI matches perfectly the theoretical curve for two channels functioning independently (Fig. 5b) suggesting that inhibition of 14-3-3 uncouples the biophysical coupling between two interacting α-subunits. Altogether, our biophysical data all strongly suggest that one α-subunit can affect the biophysical function of another α-subunit, supporting our hypothesis that Na$_v$1.5 α-subunits dimer possesses coupled gating properties. This biophysical coupling appears to be mediated by 14-3-3 interaction with the channel, as the presence of the 14-3-3 inhibitor difopein abolishes those effects.

Traditionally, WT single-channel recordings are known to often display one fast double-level event per sweep[23–25]. When recording Na$_v$1.5 single-channels from HEK293 cells, we indeed observed a fast double-level event (O$_2$), characteristic of single sodium channels (Fig. 7a and Supplementary Fig. 8), strongly suggesting a simultaneous opening and closing of a dimer. This concerted double-level opening observed with the WT channel could be due to sodium channels opening and inactivating very fast presenting with stacked simultaneous openings due to the high probability of opening at the same time. To address this issue, our single-channel recordings in Fig. 7 were performed by stepping to a less depolarized potential (−40 mV), where openings are more scattered and can occur later on, so that the precision of concerted transitions could be more easily assessed. This is also reflected at the whole-cell level (Supplementary Fig. 9A) where a delay in activation and a much slower inactivation are seen at −40 mV compared to −20 mV, the voltage at which the sodium channel peaks. This is further illustrated by summing up single-channel recordings ($n \geq 296$ sweeps) at −20 and −40 mV (Supplementary Fig. 9B) where a delay in activation and slower inactivation is also observed at −40 mV. Our single-channel recordings at −40 mV show that the sodium channel does display coupled gating properties as illustrated by mainly sharp openings that reach the double level (O$_2$), even though the openings are occurring at different time point during the pulse (Fig. 7a, b, left panels for WT alone). Interestingly, even late opening remained coupled.

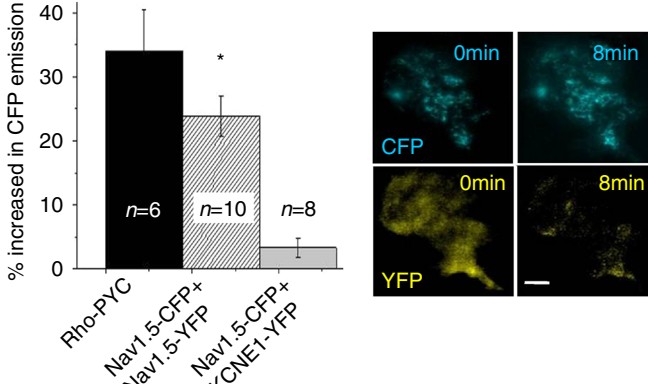

**Fig. 4** Na$_v$1.5 α-subunits still interact at the cell surface. TIRF/FRET imaging of Na$_v$1.5-CFP co-expressed with Na$_v$1.5-YFP demonstrate an increase in CFP signal following acceptor (YFP) photobleaching indicating that the two subunits are within 10 nm. Rho-PYC was used as a positive control and cells co-transfected with Na$_v$1.5-CFP+ KCNE1-YFP were used as a negative control. Scale bar = 10 μm. Data are presented as mean ± SEM, *$p < 0.05$

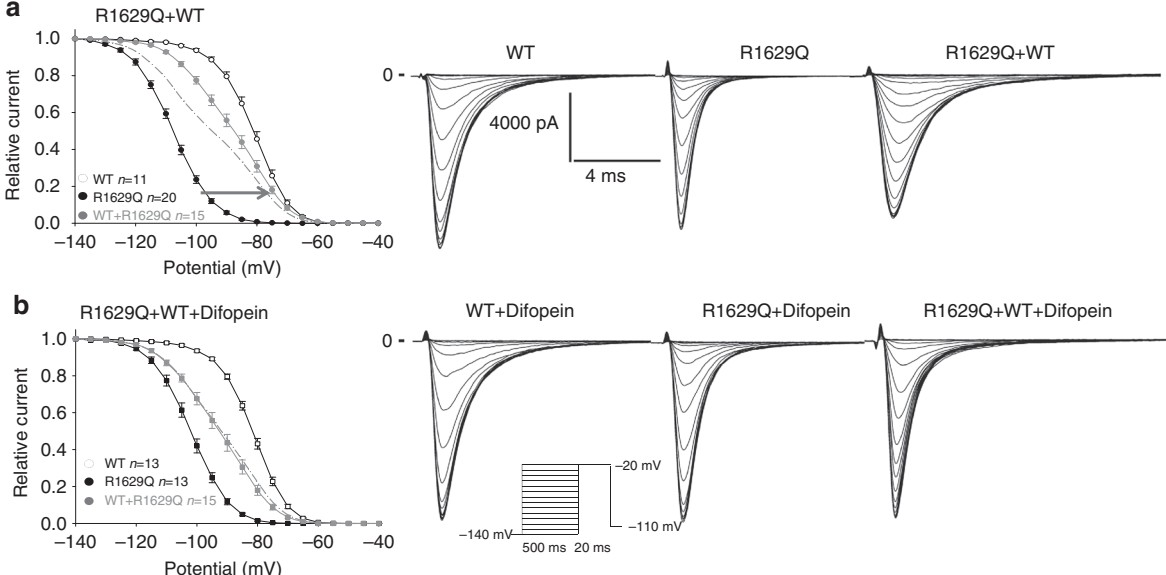

**Fig. 5** Mutant channels display biophysical coupling with WT sodium channel. **a** Steady-state inactivation for the R1629Q mutant and WT expressed in HEK293 cells. The theoretical curve (dotted line) is obtained by averaging the WT and the R1629Q mutant curves together. The dotted line corresponds to the expected theoretical curve if each channel behaved independently. Data show that co-expression of the two channels together does not behave as expected for two independent channels (theoretical curve) and therefore suggests coupling of the two channels. **b** Steady-state inactivation of WT and R1629Q mutant in HEK293 cells in presence of the 14-3-3 inhibitor difopein. Data show that the steady-state inactivation in presence of difopein now behaves as the theoretical curve for independent channels. Data are presented as mean ± SEM. Right panel displays representative current recordings for each conditions listed

Since we were able to abolish the DN effect and uncouple the biophysical properties at the whole-cell level by inhibiting 14-3-3, we hypothesized that 14-3-3 could also be mediating the single-channel coupling. Strikingly, inhibition of 14-3-3 by difopein or by introduction of the S460A mutation resulted in multiple and asynchronous single-level openings (Fig. 7a, middle and right panels respectively). This resulted in a large increase of single-level opening numbers and a drastic decrease of double-level opening numbers (Fig. 7b, middle and right panels). Interestingly, both the presence of difopein and S460A produced similar results with a dramatic decrease in double-level openings and an increase in single-level openings. Figure 7c shows that uncoupled channels can open one after the other, while the coupled channels open simultaneously. This is supported by the concept that the probability of simultaneous opening and closing at the exact same time, is highly improbable especially at −40 mV. This is also further supported by the fact that the gating probability as coupled dimer is actually sustained for a long period of time at −40 mV where we observe double openings at different time point through the pulse even if they occur later on (Fig. 7). Similar results were obtained by stepping at −20 mV, even though at this voltage, as expected, the openings are occurring earlier on and at more similar time points (Supplementary Figs. 8 and 9B). Importantly, we observed an overlap of the summation of the single current traces with and without difopein at either −40 or −20 mV (Supplementary Fig. 9B), indicating that difopein does not affect fast inactivation of the channel even though more disperse single-level openings are observed (Fig. 7 and Supplementary Fig. 8). Thus, these results demonstrate that the presence of difopein uncouples the dimers to increase the gating probability of single-channels but without affecting the inactivation kinetics (Supplementary Fig. 9C).

An additional strategy was used to overcome the low time resolution of sodium channel openings by taking advantage of an inactivation-deficient sodium channel. We used the mutant channel G1631D which displays an extremely slow inactivation

allowing us to record late events. As expected, we were able to record a larger amount of late sodium channel openings and importantly our data shows that the coupled events remained twice more probable without difopein than in presence of difopein (Supplementary Fig. 10A). This suggests that the 14-3-3 inhibitor once again uncouples the dimers even at later time points. Moreover, the total number of events represented in Supplementary Fig. 10C displays a very large number of double-level events for G1631D alone as opposed to G1631D + difopein, condition where we observed a significant loss of double-level events resulting in a major increase of single-level openings. Hence, our results with an inactivation-deficient channel, Na$_v$1.5-G1631D, again concur with a loss of simultaneous opening and closing of a pair of channels in presence of difopein. Therefore the single-channel recordings results support voltage-gated sodium channel coupling mediated through 14-3-3.

**Neuronal channels also form dimers with coupled gating**. In order to demonstrate a conserve mechanism amongst voltage-gated sodium channels, we performed crosslinking experiments, co-immunoprecipitations with 14-3-3 as well as single-channel recordings with the neuronal sodium channel isoforms Na$_v$1.1 and Na$_v$1.2. Strikingly, identical results were found; we observed the presence of a band twice the size of the monomers for the cross-linking and co-immunoprecipitations of those neuronal channels with 14-3-3 (Fig. 8a, b). We also assessed if the neuronal sodium channels displayed coupling at the single-channel level. We indeed observed a fast double-level event (O$_2$), characteristic of single sodium channels for both Na$_v$1.1 and Na$_v$1.2 expressed in HEK293 cells at either −40 mV or −20 mV (Fig. 9 and Supplementary Fig. 11, respectively), strongly suggesting a simultaneous opening and closing of a dimer also for neuronal sodium channels. More importantly, the effect of difopein on Na$_v$1.1 and Na$_v$1.2 single-channel recordings when stepping at −40 mV (Fig. 9a, b) or at −20 mV (Supplementary Fig. 11) were similar to that observed on Na$_v$1.5, i.e., loss of simultaneous gating of double-level openings for

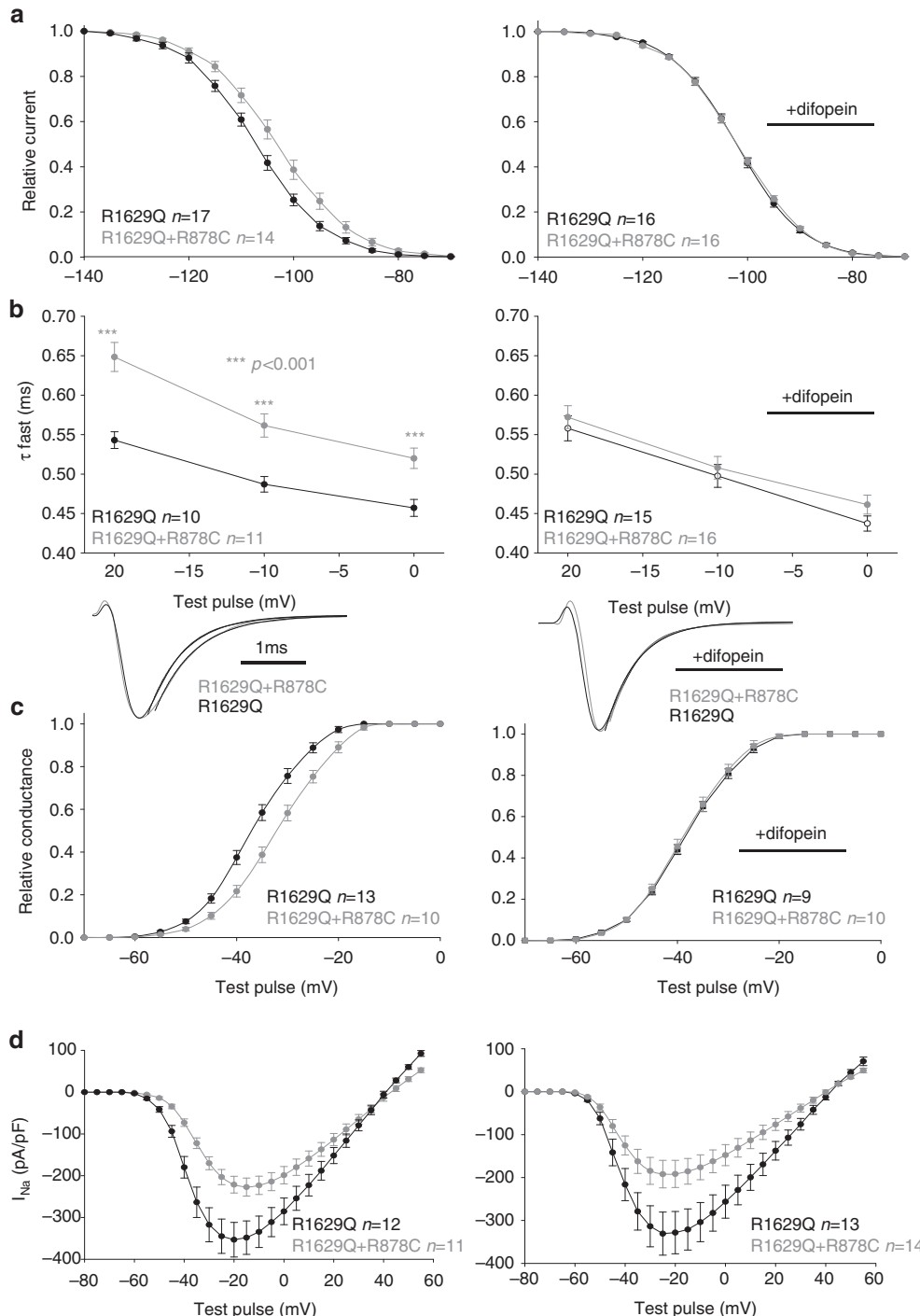

**Fig. 6** Example of biophysical coupling between R1629Q and R878C mutants when co-transfected in HEK293 cells. **a** Steady-state inactivation. **b** Time constants of inactivation measured at the listed voltages. Bottom part shows representative examples of the fit. **c** Voltage-dependent activation obtained using the conductance equation. **d** Current–voltage relationship, I/V curves. Left column without difopein, Right column in presence of difopein. Importantly, in presence of difopein we see uncoupling of the steady-state inactivation, time constants of inactivation and voltage-dependent activation. Data are presented as mean ± SEM ***$p < 0.001$

an increase of asynchronous single-level openings suggesting uncoupling of the biophysical recordings. Altogether, these results strongly support a conserved mechanism across voltage-gated sodium channel family members.

## Discussion

Voltage-gated sodium channels are crucial for proper propagation of the action potential through excitable tissues. For years, the α-

subunit, encoded by the *SCNXA* genes, was thought to be a functional monomer. However, an increasing body of literature showed inconsistency with this concept. Here, our results propose a different model of sodium channel structure, assembly and function. We demonstrate that sodium channel α-subunits not only physically interact but actually assemble, function and gate as a dimer. Importantly, we have identified the region of the channel mediating

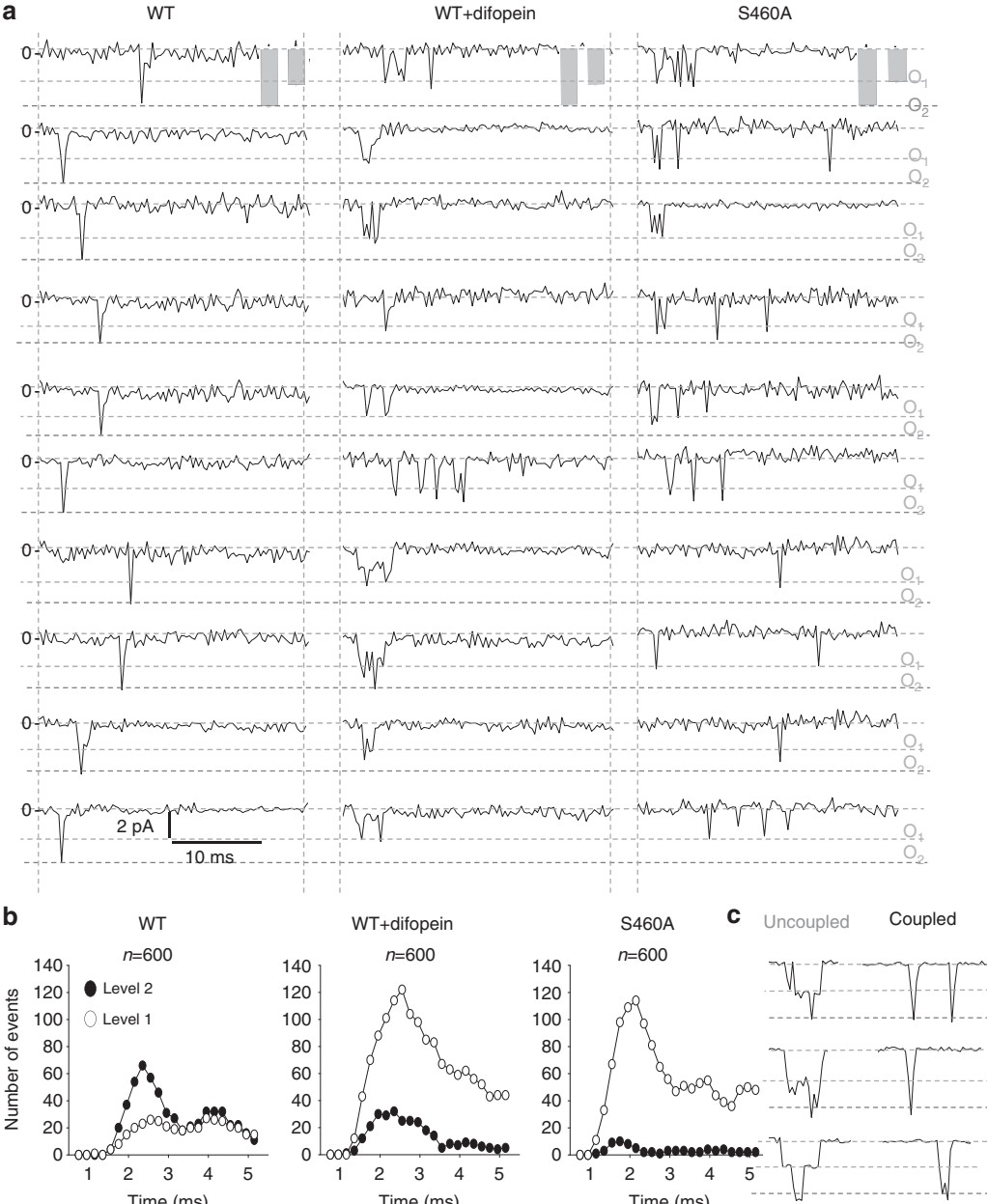

**Fig. 7** Cardiac sodium channels display coupled gating at the single-channel level which can be uncoupled by 14-3-3 inhibition. **a** Single-channel representative traces for WT, WT+ difopein and S460A at −40 mV. The overall amplitudes for single and double openings are represented as mean ± SEM in the gray bars. The dotted lines correspond to single-level openings ($O_1$) and double-level openings ($O_2$) respectively. The number of channel under a patch was estimated by the amplitude and detected by the Clampfit analysis software. **b** Number of openings in function of time. 600 hundred sweeps were recorded by depolarizing the membrane at −40 mV and the number of openings of level 1 (uncoupled) or openings of level 2 (coupled) were plotted in function of time. **c** Representative examples of uncoupled openings versus coupled openings. The uncoupled openings are non-simultaneous as opposed to the coupled opening which present a simultaneous opening of the dimer. Importantly, the loss of double-level openings results in a drastic increase of single-level openings in presence of difopein or S460A mutant and strongly supports the uncoupling of the dimers

the interaction and dimerization and elucidated that coupled gating is mediated by 14-3-3.

Unlike voltage-gated potassium channel genes which encode for one polypeptide assembling in a tetrameric complex, voltage-gated sodium channel genes (*SCNXA*) encode for an entire functional channel that has always been thought to be a monomer. However, a growing body of literature started to suggest otherwise. Indeed, different studies have shown that the common polymorphism Na$_v$1.5-H558R could partially rescue defective mutated sodium currents when present on different constructs[9,10]. In addition, several studies have also demonstrated the

presence of DN mutations found in BrS[4–6]. In fact, our previous report described a mechanism by which DN effect occurred through α-subunits interactions[4]. In the current study we investigated the α-α-subunit interaction stoichiometry and demonstrated with three different approaches (crosslinking, SiMPull experiments and binomial analysis) that contrarily to the traditional belief, sodium channels form functional dimers (Fig. 1). These results redefine our knowledge of sodium channel structure and function.

After demonstrating that sodium channel α-subunits form dimers, we established that this dimerization is mediated through

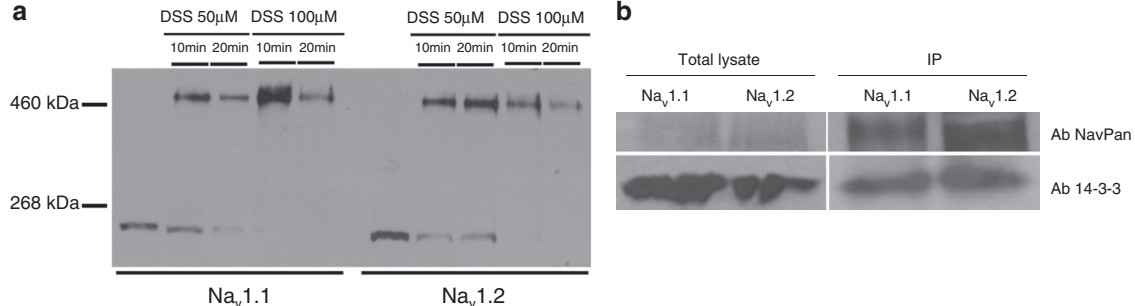

**Fig. 8** Na$_v$1.1 and Na$_v$1.2 form dimers and co-immunoprecipitate with 14-3-3. **a** Crosslinking from HEK293 cells transfected with either Na$_v$1.1 or Na$_v$1.2. Crosslinker used was DSS at either 50 or 100 μM for 10 or 20 min. **b** Co-immunoprecipitation of 14-3-3 with Na$_v$1.1 and Na$_v$1.2 in transfected HEK293 cells. Complex was co-immunoprecipitated using the sodium channel NavPan polyclonal antibody. Both sodium channel and 14-3-3 antibodies were used to reveal the interaction. Bands were shown in all conditions indicating interaction between Na$_v$1.1 or Na$_v$1.2 with 14-3-3. Full blots are presented in Supplementary Fig. 13

an interaction site found within the first intracellular loop of the α-subunit, between amino acids 493 and 517 (Fig. 2). Interestingly, Park et al.[6] displayed in a pig transgenic model a DN effect with the truncated E555X mutant channel found in a BrS family. With the assumption that DN effects are due to interactions between α-subunits, their results with this truncated channel[6] support our data that the major interaction site is found within the first 555 amino acids of the channel. Work from Gabelli et al.[26] presented a crystal structure of a partial Na$_v$1.5 C terminus fragment which they showed interacts with each other. However, the level of interaction suggested by their crystal structure might be only transitory to stabilize the closed state of the channel and thus might not be a stable interaction site to maintain the α-subunits interaction. The postulation that their structure presents a transitory configuration is in accordance with our work investigating a C-terminus truncated mutant (R1860fs12X) from an atrial fibrillation patient[17]: we demonstrated that two channels missing the C-terminus are still clearly interacting with themselves or with the WT channel even though the interaction region described by Gabelli et al. is missing, supporting that the C-terminus is not the main interaction site between channels. Importantly, our findings do not invalidate the work from Gabelli et al., since it is very likely that the dimerization of these channels will lead to an interaction between the C-terminal regions, however our results suggest that the C-terminal region is not the main region mediating the dimerization and coupled gating. More recently, Shen et al.[27], crystalized the first Eukaryotic voltage-gated sodium channel, Na$_v$PaS; this channel is conserved between 36–43% with Na$_v$1.1 to Na$_v$1.9 family members, mainly within the transmembrane segments, with a considerably shorter domain I-II linker compared to mammalian channels. Interestingly, the alignment of Na$_v$1.5 and Na$_v$PaS, showed that the entire region that we identified to mediate dimerization and the 14-3-3 binding is absent in this Na$_v$PaS channel which would contribute to the fact that they did not observe dimerization of this channel. Interestingly, crystal structure for the sodium channel accessory subunit β3 shows it interacts with the cardiac sodium channel in a trimeric complex that can oligomerize, instead of the expected canonical heterotrimeric complex containing two non-identical β subunits interacting with one α-subunit[28]. In this study, the authors also showed that β3 subunits can bind to more than one site on the Na$_v$1.5 α-subunit resulting in α-subunit oligomers, further supporting our findings.

The previously reported 14-3-3 interaction site with the channel was located between amino acids 417–467[19]. Surprisingly, our co-immunoprecipitation showed a clear interaction of 14-3-3 with the 517–555 region (Fig. 2). These data suggest that

we unveiled a second 14-3-3 interaction site just downstream of the α–α subunits interaction region (Fig. 2a, e). A similar 'double' 14-3-3 interaction region has been recently reported for Ca$_v$2.2 channel[29]. One of the two 14-3-3 binding sites found on Ca$_v$2.2 contains an ER retention signal regulated by interactions with 14-3-3 proteins which allows Ca$_v$2.2 to exit the ER. Interestingly, the other interaction site contains the phosphoserine that would regulate Ca$_v$2.2 biophysical properties. Our present data support such a mechanism for Na$_v$1.5, where phosphoserine S460 is involved in biophysical coupling of Na$_v$1.5 α-subunits mediated by 14-3-3. As observed from the co-immunoprecipitations in Fig. 2c, d, the first interaction site of 14-3-3 is lost but the interaction between α-subunits remains. This suggests that the dimerization site between α-subunits is independent of 14-3-3 interaction.

Multiple evidences exist at the single-channel level that support the coupled gating of voltage-gated sodium channels[23,24]. Aldrich et al.[23] reported that for single-channel recordings, they observed a greater likelihood for even numbers of channels under the patch. In a study by Undrovinas et al.[25] it was illustrated that under the presence of lysophosphatidylcholine (an ischemic metabolite), 2–3 synchronized channel openings were observed, while no single opening were reported on an amplitude histogram. Importantly, Naundorf et al.[1] have demonstrated that the main characteristics for the action potential initiation dynamics of cortical neurons would be better supported through a model including a cooperative activation of sodium channels. The Hodgkin–Huxley style models used failed to explain the sharp action potential initiation and actually they proposed that cooperative opening of sodium channels would be needed. Their model is consistent with our findings demonstrating that sodium channel α-subunits form a dimer that displays coupled gating properties. Our present work provides a mechanism for this cooperativity of sodium channel α-subunits mediated by the action of 14-3-3 on sodium channels. Furthermore, a recent study suggested that axonal impulse benefits from sodium channel clusterization which could have consequences in multiple diseases[30]. The dimerization of voltage-gated sodium channels that we have unveiled could be beneficial to target and concentrate channels in close area such as node of Ranvier in neurons and crest of the T-tubules and intercalated discs for cardiomyocytes where small clusters of Na$_v$1.5 have been demonstrated[31].

In conclusion, this study not only shifts paradigms in regard to sodium channel assembly and structure, but also suggests a strategy to impair DN effect and/or coupling exerted by sodium channel mutants found in different channelopathies. Considering the crucial role sodium channels play in most excitable cells, their

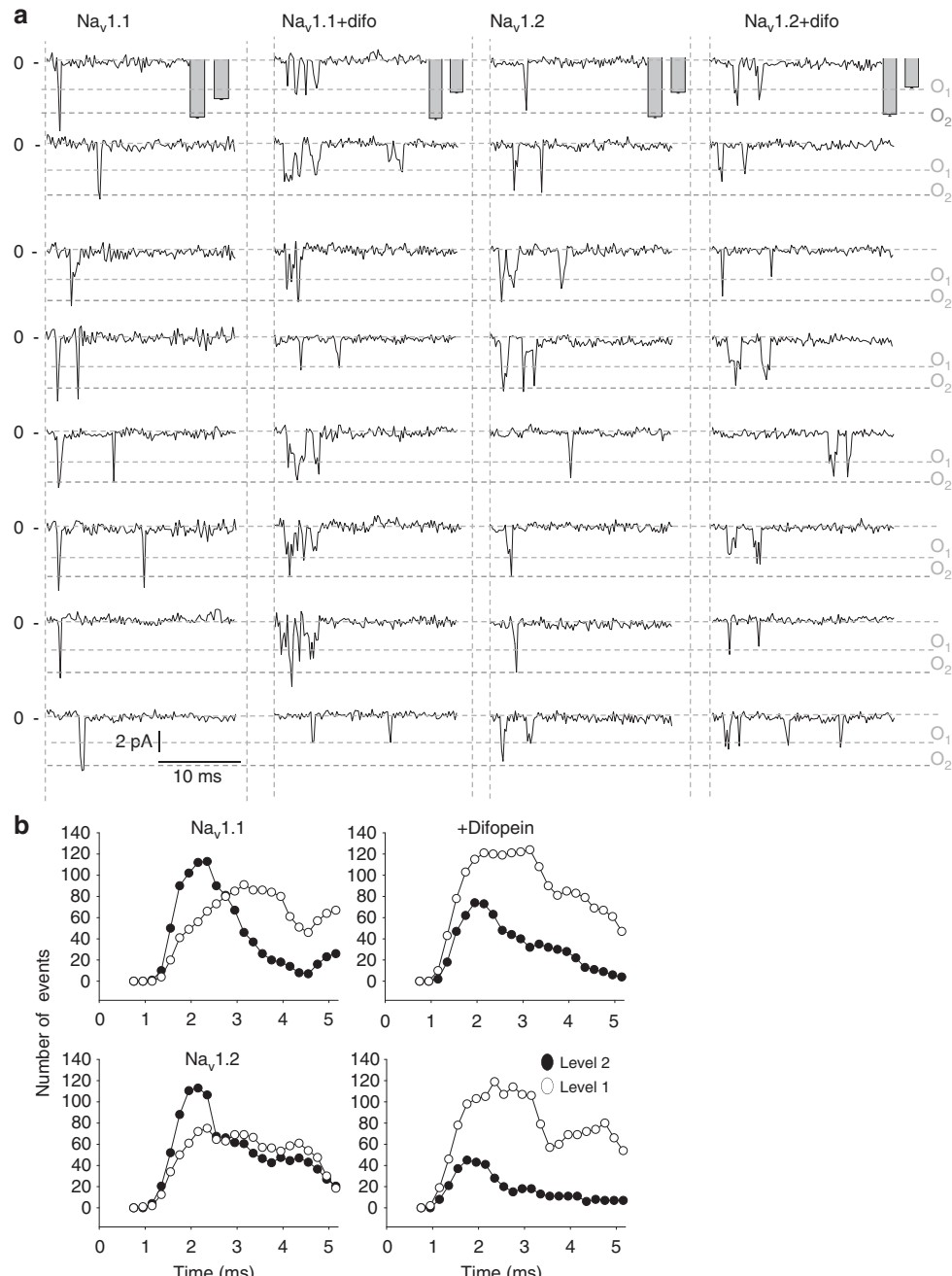

**Fig. 9** Neuronal sodium channel, Na$_v$1.1 and Na$_v$1.2 also display coupling at the single-channel level mediated through 14-3-3. **a** Representative traces of single-channel recordings at −40 mV for Na$_v$1.1 and Na$_v$1.2 expressed in HEK293 cells with or without difopein. The overall amplitudes for single and double openings are represented as mean ± SEM in the gray bars. The dotted lines correspond to single-level openings (O$_1$) and double-level openings (O$_2$), respectively. The number of channels under a patch was estimated by the amplitude and detected by the Clampfit analysis software. **b** Number of openings at the single or double level in function of time. 400 hundred sweeps were recorded by depolarizing the membrane to −40 mV. The number of openings at level 1 (single or uncoupled dimer) or openings of level 2 (coupled dimer) were plotted in function of time

key role on the evolution of excitable tissues[24,25] and their involvement in a whole host of diseases from cardiac arrhythmias to epilepsy, myotonia, paralysis, erythromelalgia, and ataxia, our results have tremendous implications allowing us to explain the mechanisms behind unexpected phenotypes in sodium channelopathies[32,33].

## Methods

**SCN5A cDNA cloning and mutagenesis**. The following plasmids were used in this study: pcDNA3.1-GFP-*SCN5A* (N-terminal-GFP)[4], pcDNA3.1-HA-*SCN5A* (gift from Dr. Peter Mohler, Columbus, OH, USA), pGFP-Ires-*SCN5A*, pEYFP-N3-

*SCN5A* (C-terminal-YFP) and pECFP-N3-*SCN5A* (C-terminal-CFP)[9]. The truncated GFP-fused channels were subcloned into the plasmid pcDNA3.1. Control experiments were performed with pcDNA3.1–YFP. The YFP fused Difopein plasmid was pEYFP-C1-difopein[34]. Na$_v$1.5 mutants were introduced by site-directed mutagenesis (QuikChange II XL Site-Directed Mutagenesis Kit, Stratagene) and validated by sequencing. All primers used in this study are listed in Supplementary Table 2.

**Cell culture and transfection**. HEK293 cells obtained through ATCC were cultured in DMEM supplemented with 10% heat-inactivated fetal bovine serum and 1% penicillin/streptomycin. All transfections were done using Polyfect (Qiagen). For the electrophysiological experiments, HEK293 cells are transfected with the

constructs of interest in 35-mm well dishes, using 0.3 µg of plasmids per 35 mm dish, to avoid saturation of the currents. To mimic the heterozygous state, cells were co-transfected with 0.15 µg of each studied constructs. For the binomial analysis while we still maintained a total amount of 0.3 µg of cDNA for each transfection, the ratio of mutant to WT cDNA varied with an increasing amount of mutant cDNA from 0 to 100% while reducing the WT amount.

For co-immunoprecipitation of Na$_v$1.5 α-subunits, HEK293 cells underwent co-transfection using 1 µg of pcDNA3.1-HA-SCN5A WT and 1 µg pcDNA3.1-GFP-SCN5A WT or mutants in 35 mm dish. For co-immunoprecipitations with Na$_v$1.1 and Na$_v$1.2, HEK293 stably expressing the channels were used (generous gift from Dr. Alfred L. George).

**Solutions for Electrophysiological recordings**. HEK293 cells were trypsinized 36 h post-transfection. Cells expressing GFP were patch-clamped. For whole-cell current recordings HEK293 cells were bathed in an extracellular Tyrode solution containing (in mM): 150 NaCl, 2 KCl, 1 MgCl$_2$, 1.5 CaCl$_2$, 1 NaH$_2$PO$_4$, 10 glucose, 10 HEPES, pH 7.4 (NaOH). Patch pipette medium was (in mM): 35 NaCl, 105 CsF, 2 MgCl$_2$, 10 EGTA, 10 HEPES, adjusted to pH 7.4 with CsOH. Extracellular tyrode solution for iCells (Cellular Dynamics Inc, Madison, WI) contained (in mM): NaCl 25, N-methyl D-glucamine 105, CsCl 5.4, MgCl$_2$ 1.8, CaCl$_2$ 2, glucose 10, HEPES 10, nisodipine 1.10$^{-3}$ to block calcium current, pH adjusted to 7.3. Patch pipette medium was (in mM): CsMES 130, TEA Cl 20, MgCl$_2$ 1, EGTA 10, HEPES 10, Mg-ATP 4, pH adjusted to 7.3. For Cell-attached single-channel recordings the pipette contained (in mM): 280 NaCl, 5.4 KCl, 1 MgCl$_2$, 2, CaCl$_2$, 10 glucose, 10 HEPES (pH 7.3). To set the resting membrane potential to 0 mV a high potassium bath solution was used: 150 mM KCl, 10 mM HEPES, pH 7.4.

**Electrophysiological recordings**. Electrophysiological recordings were performed at room temperature using the patch-clamp technique under either the whole-cell or cell-attach configuration using an Axopatch 200A amplifier (Axon Instruments, CA, USA). For whole-cell experiments, patch pipettes (Corning Kovar Sealing code 7052, WPI) with resistances between 1.9 and 2.5 MΩ were used. For cell-attached single-channel recordings, sylgard was used to reduce capacitive transients. Pipettes had resistance of 10–15 MΩ. Filtering at 5 kHz was used (−3 dB, 8-pole low-pass Bessel filter) and currents were digitized at 30 kHz (NI PCI-6251, National Instruments, Austin, TX, USA). Data acquisition was made with pClamp 10 and Clampfit (Axon Instruments) was used for analysis.

Current–voltage relationships (I/V curves) were measured by voltage steps of −100 to +60 mV in 5 or 10 mV increments for 50 ms with a holding potential of −120 mV. Steady-state inactivation −V$_m$ was obtained by holding the cells at −120 mV and then giving a pre-pulse of 2 s from −140 to −30 mV in 5 mV steps followed by a −20 mV test pulse of 50 ms. Single-channel recordings were obtained from a protocol of 100 sweeps at 0.2 Hz, depolarizing at −40 mV or −20 mV from a holding potential of −100 mV.

Activation and SSI curves were fitted to a Boltzmann equation:

$$Y = 1 / \left\{ 1 + \exp\left[ -\left( V_m - V_{1/2} \right) / k \right] \right\},$$

where $V_m$ is the membrane potential, $V_{1/2}$ is the potential for half-activation or half-availability, and $k$ is the inverse slope factor. $\tau_f$ and $\tau_s$ are fast and slow time constants of I$_{Na}$ inactivation. Single-channel events are recorded from cells with a minimum of channels under the patch in order to ensure single-channel recordings. The single-channel conductance was determined by recording over a thousand sweeps in each condition at 1 kHz frequency and no additional filtration was applied. The Clampfit analysis software was used to determine single (level 1) and second (level 2) level events after subtraction of the pipette capacitance. This allowed us to determine open level 1 and open level 2. After establishing these two open levels, we segmented the recorded traces into 0.2 ms segments, which allowed us to capture and tally closed, open level 1 or open level 2 events for each 0.2 ms segment. Gating was considered simultaneous or "coupled" when it reached the second level within 0.4 ms (two segments) see Fig. 6c showing the difference between coupling or simultaneous as opposed to uncoupled stacking or non-simultaneous.

The use of high external potassium to zero the resting membrane potential is a standard approach for single-channel recordings. The high potassium solution approximately zeroes the resting membrane potential of the overall cell, hence the voltage between the reference electrode in the bath and the measurement electrode in the pipette is equal or close to the voltage applied by the amplifier in voltage-clamp mode. Thus, to achieve a holding potential of −120 mV we apply +120 mV and to have a step pulse of −40 mV we apply +40 mV.

**Binomial analysis**. HEK293 cells are transfected with various ratios of WT and mutant cDNA. Current–voltage relationships are then recorded. Peak current densities at −20 mV are normalized to WT alone (or S460A) and plotted for each ratios. The theoretical curves are calculated using either a binomial or trinomial equation considering the probability of how the different subunits can assemble in the oligomer. For example for a dimer stoichiometry there are four possibilities: a dimer of WT molecule, a dimer of mutant molecules, but also the association of mutant/WT, and WT/mutant. We can model them as follows:

For the DN theoretical curves, we can model them as follows:

For a dimer DN theoretical curve: $y = (1 - x)^2$ where
$y$ = normalized peak current density of I/V curve (normalized to 100% functional subunit)
$x$ = ratio of DN plasmid transfected (from 0 to 1)
This equation assumes random assembly of dimeric subunits such that only two functional subunits together produce current.

However, if the two channels, WT and mutant, are independent from each other (not dimers), the current would decrease in a linear manner with Relative I$_{Na}$ = $y = (1 - x)$, as we observed when we uncoupled the dimer in presence of difopein or the S460A mutant. This corresponds to the monomeric theoretical curve.

For the trimer DN theoretical curve the equation is:
$y = (1-x)^3$ with the same $y$ and $x$ as above.

**Protein extraction**. Protein extraction was performed 48 h post-transfection of HEK293 cells with different sodium channel constructs. Cells were rinsed with PBS and lysed for 30 min on ice in lysis buffer (20 mM HEPES, pH 7.4, 150 mM NaCl, 110 mM K-Acetate, pH 7.4, 1 mM MgCl$_2$, 0.1 µM CaCl$_2$ 1% Triton X-100, 1 mM PMSF, 0.7 µg.ml$^{-1}$ Pepstatin and complete protease inhibitor cocktail were added to the lysis buffer (Roche, Germany). The soluble fractions centrifuged at 14, 000 g (4 °C) for 4 min, were then used for the experiments. Protein concentrations were measured in duplicate using a Bradford assay with a BSA standard curve. Membrane protein isolation for the co-immunoprecipitation experiments was performed as follow: Cells were washed 3 times in 1X PBS. Cells were resuspended in 3 volumes of 0.3 M sucrose/10 mM sodium phosphate plus AEBSF and complete protease inhibitor cocktail. Cells are sonicated and then incubated on ice for 30 min with vortex every 5 min. Cells are pelleted by centrifugation at 2800 r.p.m. for 10 min at 4 °C to separate nuclei and debris. Supernatant is collected and centrifuged at 8800 r.p.m. for 10 min. Supernatant is collected and centrifuged at 18,000 r.p.m. for 60 min at 4 °C. The pellet contains the membrane proteins and is resuspended in lysis buffer and incubated on ice for 10 min with vortex every 2 min. Solution is then centrifuged at 13,000 r.p.m. for 5 min at 4 °C. The supernatant contains the soluble membrane proteins[35].

**Co-immunoprecipitation**. Magnetic Dynabeads (Dynal, Norway) were washed twice with 25 mM citric acid, 50 mM Na$_2$HPO$_4$, pH 5, incubated with the mouse anti-GFP(Clontech, JL-8) or Rat anti-HA (Roche), or pan 14-3-3(H-8) from Santa Cruz Biotechnology antibodies, for 2 h at room temperature, washed three times again with 25 mM citric acid, 50 mM Na$_2$HPO$_4$, Tween 0.1%, and incubated with the pre-cleared lysates samples (total 400 mg protein) while rotating overnight at 4 °C. Beads were washed three times in lysis buffer and then proteins were eluted with the Laemmli sample buffer at 37 °C for 30 min with agitation, and analyzed by Western blot.

**Western blot**. 10% acrylamide SDS-PAGE gels are ran. Proteins are then transferred from the gels to a nitrocellulose membrane and the membrane is then incubated with the different primary antibodies: mouse anti-GFP (1:1000 Clontech, JL-8), Rat anti-HA (1:1000, Roche), the Rabbit NavPan sodium channel polyclonal antibody (1:500, Millipore), actin (Sigma-Aldrich Cat# A4700, 1:1000), hERG[13] (1:2000) and transferrin (1:1000, Sigma-Aldrich Cat#GW20009F). Bound antibodies were detected using horseradish peroxidase-conjugated goat secondary antibodies (1:5000, Santa Cruz Biotechnology), and protein signals were visualized using the SuperSignal West Dura Extended Duration Substrate (Pierce).

**Crosslinking experiments**. Protein crosslinking experiments were performed using DSS (disuccinimidyl suberate, Thermo Fisher Scientific), a noncleavable and membrane permeable crosslinker according to the manufacturer's instructions. Briefly, crosslinking of Nav1.5, Nav1.1, Nav1.2, hERG, and transferrin expressed in HEK293 cells, was performed prior to cell lysis to allow identification of the interaction between sodium channel α-subunits. Crosslinking incubation time varied between 10 and 20 min. As negative controls, samples that were exposed to all the same solutions and treatment minus the crosslinker were used. Those samples are labeled DMSO/no DSS on the different gels. After protein isolation, western blots were performed (as above). To assess size, the Hi-Mark Pre-Stained protein ladder was used (Thermo Fisher Scientific).

**TIRF/FRET**. HEK293 cells were plated on glass coverslips and transfected with 0.3 µg pECFP-N3-SCN5A and 0.3 µg pcDNA3.1-YFP-SCN5A. Total internal reflection fluorescence (TIRF) microscopy and fluorescence resonance energy transfer (FRET) experiments proceeded on an Olympus IX71 Inverted Microscope with FRET capabilities[11] supplemented with a two-line Olympus TIRFM system using a 60× TIRFM objective lens[36]. Recordings were obtained with an Hamamatsu ORCA-ER charge-coupled device (12 bit) under the control of the SLIDEBOOK software (Intelligent Imaging Innovations, Denver, CO). Images were analyzed with ImageJ using the FRETCALC plugin. FRETc was calculated as FRET − (a×CFP)−(b×YFP) where the crosstalk coefficient are $a = 0.29$ and $b = 0.04$.

**SiMPull**. Six-well plates of HEK293 cells were transfected with Polyfect as described above for patch-clamp. 0.4 µg of pcDNA3.1-GFP-Na$_v$1.5, pCLC3sGFP

(Addgene) or pcDNA3.1-Kv4.3-GFP were used. Cells were harvested at 24 h post-transfection and lysed using a CHAPS buffer (0.4% CHAPS, 20 mM Tris-HCl pH 8.0, 150 mM NaCl, 0.5 mM TCEP with protease inhibitor cocktail). Anti-GFP antibodies were immobilized on a passivated glass coverslips and lysates containing GFP-tagged channels were perfused over them. Using video software and TIRF imaging, photobleaching steps were counted to quantify the number of fluor-ophores present in each complex, thus allowing direct measurement of stoichiometry[37].

**Human derived cardiomyocytes**. Human-induced pluripotent stem cell derived cardiomyocytes (iCell Cardiomyocytes; Cellular Dynamics International) were maintained in the provided iCell Cardiomyocytes Maintenance Medium (Cellular Dynamics International) under 93% humidified air and 7% $CO_2$ at 37 °C. 20,000–40,000 cells were plated on glass coverslips coated with 0.1% gelatin and used for the patch-clamp experiments[35].

**Statistical analysis**. Results are illustrated as means ± SEM. Statistical significance was determined using SigmaPlot software with either the Student's t-test or ANOVA, as applicable. Significance was obtained for $P < 0.05$. Groups were tested for variances and were all considered of equal variances. For most results, data presented correspond to representative examples of a minimum of three experiments. We determined the group size required based on estimates of variance and minimum detectable differences between groups according to our previous recordings in HEK293 cells and iCells.

**Data availability**. The datasets generated in this manuscript are available from the corresponding authors based on reasonable request.

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

## Acknowledgements

This work was supported by an American Heart Association Scientist Development Grant (0635295N) (I.D.), NIH R01 (HL094450) (I.D.), an American Heart Association Pre-Doctoral Fellowship from the Great Rivers Affiliate 0815479D (K.S.), an American Heart Association Pre-Doctoral Fellowship from the Great Rivers 12PRE11940047 (M. H.), The Kenneth M. Rosen Fellowship in Cardiac Pacing and Electrophysiology from the Heart Rhythm Society (J.C.) and the Thailand Research Fund RSA5880015 (K.S.). We would like to thank Aurore Girardeau for her technical help.

## Author contributions

Conceptualization, J.C., K.S., E.F., and I.D.; Methodology, J.C., C.M., A.J., T.H., and I.D.; Investigation, J.C., M.H., H.L., A.J., X.W., K.S., and C.M.; Resources, I.D. and T.H.; Writing, J.C., M.H., C.M., and I.D.; Supervision, E.F., T.H., and I.D. Funding Acquisition, J.C. and I.D.

**Additional information**

**Competing interests:** The authors declare no competing financial interests.

