## [Peer Review File · Nature Communications]

Reviewers' comments:

Reviewer #1 (Remarks to the Author):

Clatot et al performed a series of complementary experiments providing evidence for a dimerization of Nav1.5 (and Nav1.1 and Nav1.2) channels, for an involvement of the 14-3-3 protein in dimerization, binding to a region around amino acid 470, and that the dimeric channel exhibits coordinated gating.

Altogether, the authors seem to be able to make a strong point, and the conclusion of a dimeric quaternary structure of Na channels would be of considerable importance in the field.

Nevertheless, several experiments are not 100% stringent and some lack important controls.

In particular:

1. Fig. 1 A, crosslinking: Full crosslinking is apparent after 1 minute. Thus the exposure for 2, 5, 10 and 20 minutes seems superfluous. It would be much nicer to see half-way cross-linking at smaller concentrations. Furthermore this experiment lacks any negative controls.
2. Fig. 1B. Everybody working with transient transfection knows that expression levels are extremely variable. The number $n > 12$ for each data point is too low in the opinion of this reviewer. Further, the experiment lacks a true negative control that could be co-expression of WT with a non-conducting mutant with normal dimerization behavior and not being dominant negative.
3. Photobleaching. From the raw traces of suppl. Figure 1 practically all dwell times of the higher level in the 2-step bleaching events of Nav1.5 are very short- at least by eye. The mean duration of these events is expected to be 1/2 of the single bleaching steps. Statistics for this should be provided.
4. What is the number of experiments in Fig. 2B, C, D?
5. A principal problem with the 14-3-3 interaction studies is that proteins from whole-cell lysates are used. Thus, a large "contamination" of proteins from the ER can be expected. It would be much stronger if proteins would be isolated from the plasma membrane.
6. Fig. 3B. What is the current density of cell transfected with WT? This is an important negative control.
7. The single channel current measurements suffer from low time resolution. Usage of inactivation deficient mutations would make this point much stronger.

Reviewer #2 (Remarks to the Author):

Clatot and co-workers report the dimeric interaction of alpha subunits of Nav1.5 and other sodium channels, in heterologous and native systems. They build on previously published evidence, but the same and other groups, that certain Na channel mutations induce a dominant negative phenotype, difficult to explain without interaction of Na channel monomers. They first perform cross-linking, single molecule pull-down and co-expression experiments. Furthermore, co-immunoprecipitation and truncation experiments, combined with pharmacological and genetic manipulation of binding of 14-3-3 to the channel identify this protein as an important factor for coupled gating. Single channel recordings show frequent occurrence of apparently simultaneous gating of two channels, which is also dependent on 14-3-3. Co-expression of mutants with altered inactivation and non-conducting ones also deviates from the predicted behavior of a mixture of

channels that gate independently. Altogether, the authors conclude that Nav1.5 (and Nav1.1 and Nav1.2) assemble as dimers that gate coordinately.

As the authors state, this had been proposed already in the very early times of patch clamping, most explicitly by Iwasa et al., 1986, ref 23. The novelty of this manuscript would reside in the need for dimerization for function, rather than the possibility that dimers with coupled gating exist. The experiments are convincing with respect to an interaction between subunits, although some clarification in certain aspects would be required.

Specifically:

1-One would interpret that dimeric, coupled channels are the common stoichiometry of sodium channels. This is for example the assumption for the binomial analysis. There is however extensive literature showing single sodium channels acting as independent entities with the expected unitary conductance, what would be the reason for this?

2-Cross-linking gives rise to a band size of roughly twice that of the monomer (in the absence of cross-linking). This is however surprising, as it implies that the only interaction is between two alpha subunits, without other accessory subunits and interaction proteins, including 14-3-3.

3-GFP itself has tendency to dimerize. It is not fully clear which version of GFP was used for SiMPull experiments?

4-Why does the S460A mutant reduce the interaction, if the "relevant" 14-3-3 binding site is still present?

5-From IP experiments it seems that all channels are bound to 14-3-3 (the intensity of the co-IP is the same as the intensity in the extract), is this correct?

6-493X seems to preserve a certain degree of interaction. Could this be interpreted as some region between 470 and 493 also participating of the interaction?

Reviewer #3 (Remarks to the Author):

Clatot et al. performed a series of assays and experiments to demonstrate that voltage-gated sodium channel alpha-subunits (Nav1.5) act in a dimer configuration. Additionally, the authors presented meaningful results revealing the mechanism of subunit interaction. Previously, it was expected that alpha-subunits of sodium channels function as monomers. This article provides specific information that these subunits not only interact with each other, but also assemble, function and gate as dimers. This interaction is supported/mediated by 14-3-3 protein and is conserved amongst mammalian sodium channels, which could lead to new approaches for treating/preventing sodium channelopathies.

In order to investigate the stoichiometry of the complex formed by the sodium channel alpha-subunits, a classical ensemble approach of crosslinking with DSS was applied. Crosslinking provided first indications of a dimer configuration. Additionally, the SiMPull-method (single-molecule pull-down), which was introduced and established by Ha et al. (2011), confirmed the assumption of a dimer-formation. The advantage of SimPull is the visualization of individual in-vivo cellular protein complexes providing information on the complex stoichiometry and composition by means of a TIRF-microscope. For these purposes, cell lysate expressing GFP-tagged Nav1.5-subunits was immunoprecipitated on the microscope slides functionalized with anti-GFP antibodies. Photobleaching-steps analysis revealed nearly a 1:1 distribution of 1- and 2-step bleaching events. Although, unlike organic dyes genetically encoded GFP-tags make sure that each tagged Nav1.5-subunit is coupled to one GFP, some fraction of the fluorescent proteins (here ca. 30 %) can be fluorescently inactive. These circumstances are well known in the community and it is due to some folding issues and low photophysical stability of FPs. One of the strengths of this manuscript with respect to the SiMPull-experiments is the comparison of distributions of other known interacting membrane proteins which leads to a consistent conclusion of a dimer formation by Nav1.5-

subunits. Additionally, SiMPull-experiments were carefully carried out and evaluated. An additional approach of binomial analysis based on the current density was used to examine the stoichiometry. Overall, the manuscript is well written and the high-quality data are presented clearly. It is convincingly shown that the single-molecule method provides useful and important biological information valuable for the community. I have no objections to publication.

Nature Communications: Voltage-Gated Sodium Channels Assemble and Gate as Dimers

We are very grateful for these insightful reviews. We are pleased that the reviewers appreciated the potential implication of the paper and we were encouraged by comments such as: *“Altogether, the authors seem to be able to make a strong point, and the conclusion of a dimeric quaternary structure of Na channels would be of considerable importance in the field.”* or *“The experiments are convincing with respect to an interaction between subunits”* and *“One of the strengths of this manuscript with respect to the SiMPull-experiments is the comparison of distributions of other known interacting membrane proteins which leads to a consistent conclusion of a dimer formation by Nav1.5-subunits. Additionally, SiMPull-experiments were carefully carried out and evaluated. Overall, the manuscript is well written and the high-quality data are presented clearly. It is convincingly shown that the single-molecule method provides useful and important biological information valuable for the community.”* We also appreciate and greatly value the constructive comments from the reviewers and thank the reviewers for their suggestions which we believe have improved the scientific content of the manuscript.

We are excited to say that we believe we were able to answer all of the reviewers' concerns and performed all of the requested experiments. Therefore, we hope that you will find answers to your questions and concerns in our new set of experiments. Responding to the major critique from all reviewers:

1. We now provide crosslinking control experiments and have redone the crosslinking with lower concentrations of DSS. The results further support our conclusion that sodium channels form dimers.
2. Additionally, as cleverly suggested by the reviewers, single channel recording experiments of an inactivating deficient channel have been performed which enabled us to observe coupled gating at later time points where the openings are scattered and not stacked, circumventing the low time resolution issue. These results once again strengthen our conclusion that sodium channels form dimers with coupled gating.

The content of the manuscript has been modified and improved in accordance with your suggestions and the modified text is in displayed in blue.

Please also find below each of the reviewers' comments in **bold** followed by our responses.

Reviewer #1:

Clatot et al performed a series of complementary experiments providing evidence for a dimerization of Nav1.5 (and Nav1.1 and Nav1.2) channels, for an involvement of the 14-3-3 protein in dimerization, binding to a region around amino acid 470, and that the dimeric channel exhibits coordinated gating. Altogether, the authors seem to be able to make a strong point, and the conclusion of a dimeric quaternary structure of Na channels would be of considerable importance in the field.

We thank the reviewer for her/his acknowledgement that our findings would be of considerable importance to the field.

Nevertheless, several experiments are not 100% stringent and some lack important controls.

1.Fig. 1 A, crosslinking: Full crosslinking is apparent after 1 minute. Thus the exposure for 2, 5, 10 and 20 minutes seems superfluous. It would be much nicer to see half-way cross-linking at smaller concentrations. Furthermore this experiment lacks any negative controls.

This is an excellent observation. We have therefore redone the crosslinking experiments for Nav_v1.5, but now used smaller concentrations of DSS for 20 minutes (10μM, 25μM or 50μM) instead of the 100μM previously utilized. Using these concentrations, we now observe small amount of crosslinking with lower concentrations (10μM or 25μM) and a more predominant crosslinked dimer band with 50μM. We have replaced the previous Fig.1A with these new results.

We apologize for not originally providing negative controls for the crosslinking experiments. We have now performed several controls. First, in our new crosslinking experiments we now include a sample that was exposed to all the same solutions and treatment minus the crosslinker. Those samples are labeled DMSO/ no DSS on the different gels. Second, we performed crosslinking experiments on transferrin, a protein that is not known to oligomerize. We can see for this control that only one band at the monomeric size of about 77kDa is

observed and no higher bands are present. This is now included in Supplemental Fig. 1A. Finally, we co-expressed Na_v1.5 with the potassium channel hERG, a protein with which it does not interact. We then performed crosslinking and revealed for either hERG or the sodium channel. When probing the membrane for the sodium channel, we obtained similar results as in Fig. 1A, with bands at similar expected sizes for a monomer and a dimer in presence of DSS, confirming no interaction with hERG. When revealing the western blot for hERG, in presence of DSS we obtained 3 different bands with sizes expected for either a monomer, dimer, or tetramer as previously reported from our group (Wang et al., Mol Pharmacol. 2009 Apr;75(4):927-37). This result provides an additional control for the sodium channel dimerization status. Importantly, this experiment in presence of hERG also provides us with a positive control using a protein of known oligomerization status which further validates our crosslinking approach. Therefore, these new negative and positive controls for the crosslinking experiments further support our conclusion that sodium channels form dimers.

2. Fig. 1C. Everybody working with transient transfection knows that expression levels are extremely variable. The number n>12 for each data point is too low in the opinion of this reviewer. Further, the experiment lacks a true negative control that could be co-expression of WT with a non-conducting mutant with normal dimerization behavior and not being dominant negative.

We sincerely thank the reviewer for this comment and understand the concerns. This is a valid point that we always come across when performing transfection and co-transfection in expression systems. We have extensive experience with this issue based on our previous work and have once again taken all the appropriate steps to ensure that the channels expression is as little variable as possible. From our experience, we have seen that it is important to use low amount of cDNA and have found that in ranges between 0.15µg to 0.6µg we observe changes in current densities that are proportional to the amount of cDNA transfected and our current densities are usually quite tight. We also always keep the amount of cDNA transfected constant across the different conditions and always perform the experiments that will be compared on the same day as controls. We show in supplemental figure 3, representative traces, and I/V curves for the different ratios for the binomial analysis so that the current density measurement can be appreciated as a valid reflection of the level of channel expressed. Therefore while we agree that this is a potential area of concern, we are confident that this is an issue that we have carefully addressed over the years and feel confident that the level of expression of the different WT:mutant ratios and the current density are directly correlated. Obviously, large number of n are also required to validate these results over multiple transfections and various conditions. Our revised manuscript now includes the n values in Supplemental Table 1 to keep the figures as clear as possible. Please note that the number of experiments (n) for each condition are far superior than 12; an n of 12 was the minimum number of cells patched for the cells expressing the mutant channels alone which leads to no current. We apologize for the confusion.

We thank the reviewer for the excellent suggestion of co-expressing the WT with a non-conducting mutant with normal dimerization behavior which is not dominant-negative (DN). We performed an additional binomial analysis by co-expressing the WT with the trafficking but non-conducting mutant R878C, which dimerizes (as illustrated in this study in Fig. 6 due to shifts in biophysical properties in presence of this non-conducting mutant) but does not exert a dominant negative effect as illustrated in our previous work (Clatot et al., Cardiovasc Res. 2012 Oct 1;96(1):53-63). As observed in our previous study and in this paper, DN-mutants all lead to a 75% decrease in I_{Na} when both WT and DN-mutants are expressed at a 50:50 ratio suggesting a dimer. However, when we co-expressed the non-conducting, non DN-mutant R878C with WT as suggested by the reviewer, this led to a reduction of only 50%. We performed the full binomial analysis using this mutant and the WT and demonstrate that the reduction in I_{Na} current density follows a linear decrease which reinforces our hypothesis that trafficking efficient channels are less likely to exert a DN-effect. This new control is now included in Supplemental Fig. 3C.

3. Photobleaching. From the raw traces of suppl. Figure 1 practically all dwell times of the higher level in the 2-step bleaching events of Na_v1.5 are very short- at least by eye. The mean duration of these events is expected to be 1/2 of the single bleaching steps. Statistics for this should be provided.

The reviewer raises an interesting point. Photobleaching of fluorophores is a stochastic process and for a dimer, in principle, both GFP molecules should bleach independent of one another. We provide new data in

Suppl. Fig. 2D showing that for molecules bleaching in 2 distinguishable steps, the mean photobleaching time for the first step is 0.9 ± 0.2 s while that for the complete bleaching (total time till the second step) is 4.5 ± 2.0 s. In contrast, the mean bleaching time for the molecules bleaching in one-step only is 1.4 ± 0.2 s. The short photobleaching time for the first bleaching step is indeed expected, when the bleaching of two fluorophores is stochastic and independent of one another. We explain these results using the analogy of dice. Let's assume that the value on dice face corresponds to the bleaching time of a fluorophore. The expected value for a dice roll is 3.5 (average of 1-6). A dimer is equivalent to two dices being rolled out together. The earlier bleaching event is analogous to the lower of the two dice values. The expected value for lower dice roll (~ 2.52) is lower than the expected value when only one dice is rolled (3.5). The second bleaching event is equivalent to the higher of the two dice values and is expected to be higher (4.5) than the single dice roll.

4. What is the number of experiments in Fig. 2B, C, D?

The number of experiments for Fig. 2B are listed within each bar corresponding to each of the different channels studied. The co-immunoprecipitations in Fig. 2C were repeated 4 times and the co-immunoprecipitation in Fig. 2D were repeated 5 times. This is now listed in the figure legend.

5. A principal problem with the 14-3-3 interaction studies is that proteins from whole-cell lysates are used. Thus, a large “contamination” of proteins from the ER can be expected. It would be much stronger if proteins would be isolated from the plasma membrane.

This is a great suggestion. We have redone the co-immunoprecipitation for the 555 fragment which interacted with 14-3-3 and confirmed that the interaction still occurs in the plasma membrane. It is important to note however that a very small fraction of these truncated channels reaches the plasma membrane as can be expected for several of the DN-mutants. But nevertheless, the ones that did, do interact with 14-3-3 in the membrane. This confirms our hypothesis that 14-3-3 is interacting with the channel at the cell surface, and strengthens our findings that 14-3-3 is involved in the coupled gating of the channel. This result is now included in Fig. 2D.

6. Fig. 3B. What is the current density of cell transfected with WT? This is an important negative control.

We apologize for not including this control. Current density for cells transfected with WT is 2.7 ± 0.3 pA/pF. This is now included in Fig. 3B.

7. The single channel current measurements suffer from low time resolution. Usage of inactivation deficient mutations would make this point much stronger.

We agree with the reviewer that the use of an inactivation deficient channel would help measuring the double level opening at later time points and resolve the low time resolution. In order to answer this question, we used the mutant channel G1631D which displays an extremely slow inactivation and therefore would allow us to record late events. G1631D single channel recordings are now presented in supplemental figure 10. As expected, we were able to record a larger amount of late sodium channel openings and more importantly our data show that the coupled events remained twice more probable without difopein than in presence of difopein (Supplemental Figure 10A). This suggests that the 14-3-3 inhibitor once again uncouples the dimers even at later time points. Moreover, the total number of events represented in Supplemental Figure 10C strikingly displays a very large number of double level events for the G1631D alone as opposed to G1631D + difopein, condition where we observed a drastic loss of double level events resulting in a major increase of single level openings. Hence, our results with an inactivation deficient channel, $\text{Na}_v1.5\text{-G1631D}$ again concur with a loss of simultaneous opening and closing of a pair of channels in presence of difopein.

Reviewer #2:

Clatot and co-workers report the dimeric interaction of alpha subunits of $\text{Na}_v1.5$ and other sodium channels, in heterologous and native systems. They build on previously published evidence, but the same and other groups, that certain Na channel mutations induce a dominant negative phenotype,

difficult to explain without interaction of Na channel monomers. They first perform cross-linking, single molecule pull-down and co-expression experiments. Furthermore, co-immunoprecipitation and truncation experiments, combined with pharmacological and genetic manipulation of binding of 14-3-3 to the channel identify this protein as an important factor for coupled gating. Single channel recordings show frequent occurrence of apparently simultaneous gating of two channels, which is also dependent on 14-3-3. Co-expression of mutants with altered inactivation and non-conducting ones also deviates from the predicted behavior of a mixture of channels that gate independently. Altogether, the authors conclude that Nav1.5 (and Nav1.1 and Nav1.2) assemble as dimers that gate coordinately. As the authors state, this had been proposed already in the very early times of patch clamping, most explicitly by Iwasa et al., 1986, ref 23. The novelty of this manuscript would reside in the need for dimerization for function, rather than the possibility that dimers with coupled gating exist. The experiments are convincing with respect to an interaction between subunits, although some clarification in certain aspects would be required.

We thank the reviewer for the insightful comments and appreciate that he/she thought our experiments were convincing with respect to an interaction between subunits.

Specifically:

1. One would interpret that dimeric, coupled channels are the common stoichiometry of sodium channels. This is for example the assumption for the binomial analysis. There is however extensive literature showing single sodium channels acting as independent entities with the expected unitary conductance, what would be the reason for this?

This is a very good point. It is indeed possible to record single entities of sodium channels as we also observed in the present study, albeit at a much lower levels. We do however observe a very significant increase in single level conductance when the cells are under conditions where 14-3-3 function is inhibited (either with difopein or with S460A mutant). Indeed, from our experiments we show that the sodium channel does display coupled gating properties as illustrated by mainly double level openings, even though the openings are occurring at different time point during the pulse. Importantly though, in presence of difopein or S460A, the double level openings are decreased significantly, showing uncoupling of the channels resulting in single level openings. This was observed at different voltages, for Nav1.5, Nav1.1, Nav1.2 and even an inactivation deficient Nav1.5 mutant (G1631D). We can see by comparing the results at -20mV versus -40mV a difference in the temporal dispersion of the openings, as expected. However, the proportion of double level remains similar confirming that the channels are coupled at both voltages. The fact that at -40mV, difopein (or S460A), significantly decreases the probability of double openings during the entire time course, including late openings demonstrates uncoupling of the dimer. Therefore, we feel that the clear difference in number of single versus coupled levels channel under the different conditions support our findings of channels having a high tendency for a dimeric coupled configuration.

Hence, we don't disagree with the ability to record single sodium channel entities. But, there are multiple possibilities as to why different reports might see a predominance of single versus coupled entities. For example, it might depend on the recording conditions or systems in which the recordings were made. Indeed, the conditions could influence the 14-3-3 biophysical coupling of the channels which could explain why some groups have observed more frequent channels with unitary conductance. In fact, we know that 14-3-3 interaction with its target proteins is phosphorylation dependent, it is therefore possible that different levels of phosphorylation could influence the coupling explaining why you can obtain single sodium channel recordings. The expression system could also be a factor depending on the presence of 14-3-3 or even which 14-3-3 isoform is present.

However, one could also argue that there is abundant literature supporting the coupled single channel recordings. Indeed, while it is true that groups were able to measure sodium channel acting as independent entities, many groups also noted paired openings of sodium channels: 1. Aldrich et al "noted a tendency for even number of channels to occur within a patch for single-channel recordings", 2. Undrovinas et al. also noted 2 to 3 synchronized sodium channel openings from isolated patch clamp experiments and more recently 3. the group of Dr. Mario Delmar demonstrated in cardiomyocytes the clustering of sodium channels and that the openings were often either 0 or even numbers. 4. And finally, Naundorf *et al* have demonstrated that key

features of the initiation dynamics of cortical neurons action potentials would be better supported through a new model based on the cooperative activation of sodium channels. While they failed to explain these features using single- or double-compartment Hodgkin–Huxley style models, they suggested instead that they could arise from cooperative opening of sodium channels. Their model is consistent with our findings demonstrating that sodium channel α -subunits form a dimer that displays coupled gating properties.

2. Cross-linking gives rise to a band size of roughly twice that of the monomer (in the absence of cross-linking). This is however surprising, as it implies that the only interaction is between two alpha subunits, without other accessory subunits and interaction proteins, including 14-3-3.

This is another excellent point from the reviewer. As mentioned by reviewer 1, our crosslinking experiments were missing some important controls that could also help answer the issue brought up by reviewer 2 about the interaction with other proteins. First, in our new crosslinking experiments we now include a sample that was exposed to all the same solutions and treatment minus the crosslinker. Those samples are labeled DMSO/ no DSS on the different gels. Second, we performed the crosslinking experiments on transferrin, a protein that is not known to oligomerize but which could also interact with other proteins. Yet, we can see for this control that only one band at the monomeric size of about 77kDa is observed and no higher bands are present. This is now included in Supplemental Fig. 1A. We also co-expressed Nav1.5 with the potassium channel hERG, a protein with which it does not interact. We then performed crosslinking and revealed for either hERG or the sodium channel. When probing the membrane for the sodium channel, we obtained similar results as in Fig. 1A, with bands at similar expected sizes for a monomer and a dimer in presence of DSS, confirming no interaction with hERG. When revealing the western blot for hERG, in presence of DSS we obtained 3 different bands with sizes expected for either a monomer, dimer, or tetramer as previously reported from our group (Wang et al., Mol Pharmacol. 2009 Apr;75(4):927-37). This result further supports the lack of interaction between Nav_v1.5 and hERG and provides an additional control for the sodium channel dimerization status. Importantly, this experiment in presence of hERG also provides us with a positive control using a protein of known oligomerization status which further validates our crosslinking approach. Even though hERG is also known to interact with smaller partners and chaperones present in cells, the 3 bands obtained for hERG were at the expected sizes for a monomer, dimer and tetramer. Therefore, this hERG positive controls for the crosslinking experiments further support our conclusion that the band observed for the sodium channel is for a dimer. While it is possible that other small proteins could be crosslinked, one issue could be that it is hard to evaluate the exact size of the dimer due to the high molecular weight.

3. GFP itself has tendency to dimerize. It is not fully clear which version of GFP was used for SiMPull experiments?

The reviewer is correct; GFP has been reported to have a tendency to dimerize under certain conditions. To avoid that, one can mutate some amino acids referred to as monomerizing mutants. However, in our case we used the non-mutated GFP for all of our constructs used in the SiMPull experiments: GFP-Nav_v1.5, K_v4.3-GFP and ClC-3-GFP. Therefore, since the GFP-fused Na, K and Cl channels do not have the monomerizing mutations, we feel confident that since we are comparing channels with the same GFP construct, the contribution of weak potential basal dimerization of GFP should be the same for all of these constructs. Importantly, the fact that the distribution observed for the photobleaching steps for a known dimer, the chloride channel, behaved identically to the sodium channel further validates our findings and excludes a potential artifact of the GFP. Similarly, the results with the known tetramer potassium channel further validated the SiMPull methodology displaying a distribution of 4 photobleaching steps. Finally, while the SiMPull experiments used a fused GFP construct, it is important to emphasize that for all the other experiments used to assess the stoichiometry of the sodium channel, the channel was actually not tagged to GFP, but instead we were using an Ires construct. So, for the crosslinking, the binomial analysis and all the electrophysiology clearly the dimerization we see is not due to GFP. Therefore, we feel confident that the different controls in the SiMPull experiments and the multiple approaches used to assess the stoichiometry allow us to conclude that the sodium channel dimerizes and our results are not due to GFP dimerization.

4. Why does the S460A mutant reduce the interaction, if the “relevant” 14-3-3 binding site is still present?

The previously reported 14-3-3 interaction site with the channel was located between amino acids 417-467 (Allouis et al., 2006). Surprisingly, our co-immunoprecipitation showed a clear interaction of 14-3-3 with the 517-555 region (Fig 2). These data suggest that we unveiled a second 14-3-3 interaction site just downstream of the α - α subunits interaction region (Fig. 2A and E). A similar 'double' 14-3-3 interaction region has been recently reported for Cav2.2 channel, where 14-3-3 can interact with two different but close C-terminus regions of the channel (Liu et al., 2015). One of the two 14-3-3 binding sites found on Cav2.2 contains an ER retention signal that is subjected to regulation by 14-3-3 proteins enabling the channel to escape the ER. Interestingly, the other interaction site contains the phosphoserine that they suggest would regulate Cav2.2 biophysical properties. Our present data support such a mechanism for Nav1.5, where phosphoserine S460 is involved in biophysical coupling of Nav1.5 α -subunits mediated by 14-3-3. As you can observe from the co-immunoprecipitations in figure 2C and 2D, the first interaction site of 14-3-3 is lost but the interaction between α -subunits remains. Hence, we think that the interaction site between the two α -subunits is framed by the two 14-3-3 interaction site but independent of the interaction of 14-3-3. We apologize for not discussing this in the previous version. We have now included this in the revised version of the discussion.

5. From IP experiments it seems that all channels are bound to 14-3-3 (the intensity of the co-IP is the same as the intensity in the extract), is this correct?

This is a very good point revealed by the reviewer; the ratio between the two bands seems close to one and always very strong. However, these co-IP experiments are not quantitative and therefore it is hard to prove that all channels are bound to 14-3-3 proteins. But we believe that a large majority of the channels are linked to 14-3-3 proteins. To the best of our knowledge, 14-3-3 expression level in cardiomyocytes has been reported to be as high as 1% of the total protein expression level; therefore, these proteins represent a very large amount of proteins in the cell and potentially even more than voltage-gated sodium channels, which may allow the cells to saturate the channels with the 14-3-3 binding partner. In addition, 14-3-3 are ubiquitous and also largely present in HEK-293 cells. Hence, we believe that 14-3-3 protein is an important partner of Nav1.5, Nav1.1 and Nav1.2 as it is likely involved in both trafficking and gating regulation of these channels.

6-493X seems to preserve a certain degree of interaction. Could this be interpreted as some region between 470 and 493 also participating of the interaction?

This is correct; it is not yet understood how the different alpha-helix of the region fold to maintain the dimer together, therefore further 3-dimensional studies will be necessary to better understand the 3-Dimensional structure of the region but it is possible that 470-493 region still constitute a weaker interaction site.

Reviewer #3:

Clatot et al. performed a series of assays and experiments to demonstrate that voltage-gated sodium channel alpha-subunits (Nav1.5) act in a dimer configuration. Additionally, the authors presented meaningful results revealing the mechanism of subunit interaction. Previously, it was expected that alpha-subunits of sodium channels function as monomers. This article provides specific information that these subunits not only interact with each other, but also assemble, function and gate as dimers. This interaction is supported/mediated by 14-3-3 protein and is conserved amongst mammalian sodium channels, which could lead to new approaches for treating/preventing sodium channelopathies.

In order to investigate the stoichiometry of the complex formed by the sodium channel alpha-subunits, a classical ensemble approach of crosslinking with DSS was applied. Crosslinking provided first indications of a dimer configuration. Additionally, the SiMPull-method (single-molecule pull-down), which was introduced and established by Ha et al. (2011), confirmed the assumption of a dimer-formation. The advantage of SimPull is the visualization of individual in-vivo cellular protein complexes providing information on the complex stoichiometry and composition by means of a TIRF-microscope. For these purposes, cell lysate expressing GFP-tagged Nav1.5-subunits was immunoprecipitated on the microscope slides functionalized with anti-GFP antibodies. Photobleaching-steps analysis revealed nearly a 1:1 distribution of 1- and 2-step bleaching events. Although, unlike organic dyes genetically encoded GFP-tags make sure that each tagged Nav1.5-subunit is coupled to one GFP, some fraction of the fluorescent proteins (here ca. 30 %) can be fluorescently inactive. These circumstances are well

known in the community and it is due to some folding issues and low photophysical stability of FPs. One of the strengths of this manuscript with respect to the SiMPull-experiments is the comparison of distributions of other known interacting membrane proteins which leads to a consistent conclusion of a dimer formation by Nav1.5-subunits. Additionally, SiMPull-experiments were carefully carried out and evaluated.

An additional approach of binomial analysis based on the current density was used to examine the stoichiometry.

Overall, the manuscript is well written and the high-quality data are presented clearly. It is convincingly shown that the single-molecule method provides useful and important biological information valuable for the community. I have no objections to publication.

We sincerely thank the reviewer for the great comments and are grateful that he/she could appreciate the value and implication of our findings.

REFERENCES:

Wang L, Dennis AT, Trieu P, Charron F, Ethier N, Hebert TE, Wan X, Ficker E. Intracellular potassium stabilizes human ether-à-go-go-related gene channels for export from endoplasmic reticulum. *Mol Pharmacol*. 2009 Apr;75(4):927-37.

Clatot, J. et al. Dominant-negative effect of SCN5A N-terminal mutations through the interaction of Na(v)1.5 alpha-subunits. *Cardiovasc Res* 96, 53-63 (2012).

Aldrich, R.W., Corey, D.P. & Stevens, C.F. A reinterpretation of mammalian sodium channel gating based on single channel recording. *Nature* 306, 436-41 (1983).

Undrovinas, A.I., Fleidervish, I.A. & Makielski, J.C. Inward sodium current at resting potentials in single cardiac myocytes induced by the ischemic metabolite lysophosphatidylcholine. *Circ Res* 71, 1231-41 (1992).

Agullo-Pascual, E. et al. Super-resolution imaging reveals that loss of the C-terminus of connexin43 limits microtubule plus-end capture and Nav1.5 localization at the intercalated disc. *Cardiovasc Res* 104, 371-81 (2014).

Naundorf, B., Wolf, F. & Volgushev, M. Unique features of action potential initiation in cortical neurons. *Nature* 440, 1060-3 (2006).

Allouis, M. et al. 14-3-3 is a regulator of the cardiac voltage-gated sodium channel Nav1.5. *Circ Res* 98, 1538-46 (2006).

Liu, F., Zhou, Q., Zhou, J., Sun, H., Wang, Y., Zou, X., Feng, L., Hou, Z., Zhou, A., Zhou, Y., et al. (2015). 14-3-3tau promotes surface expression of Cav2.2 (alpha1B) Ca²⁺ channels. *The Journal of biological chemistry* 290, 2689-2698.

REVIEWERS' COMMENTS:

Reviewer #1 (Remarks to the Author):

All issues have been adequately addressed.

Reviewer #2 (Remarks to the Author):

The new version and the authors' rebuttal address satisfactorily all my concerns.